# FlatLand: Personalized Graph Federated Learning via Tailored Lorentz Space

## Abstract

Federated learning enables collaborative model training across multiple clients while preserving data privacy, but faces significant challenges when client data distributions are highly heterogeneous. This problem is particularly severe in graph federated learning, where clients possess graphs with diverse structural properties. Existing personalized federated learning (PFL) methods ignore the intrinsic geometric properties of diverse graph structures. We propose FlatLand[1], a novel personalized **F**ederated **lea**rning method that embeds different clients' data in **t**ailored **L**orentz space of hyperbolic geometry. Our key insight is that hyperbolic geometry naturally accommodates the intrinsic negative curvature prevalent in real-world graphs, while the time-like dimension in Lorentz space provides a principled way to encode client-specific heterogeneity. We develop a parameter decoupling strategy that separates heterogeneous information (captured in time-like parameters) from common knowledge (preserved in space-like parameters), enabling direct aggregation without requiring client similarity estimation and extra calculation modules. Empirical results on various federated graph learning tasks demonstrate that FlatLand achieves superior performance, particularly in low-dimensional settings. The code is available at the anonymous repository.

## 1 Introduction

Federated learning (FL) has emerged as a paradigm that enables collaborative machine learning across multiple clients while preserving data privacy. Traditional FL struggles with data heterogeneity, as one model cannot satisfy diverse local requirements (Tan et al., 2022). This challenge is magnified in graph federated learning, where complex topology yields pronounced structural heterogeneity across clients (Xie et al., 2021; Tan et al., 2023). In severe cases, federated learning may even underperform local training (Baek et al., 2023). To address heterogeneity, Personalized federated learning (PFL) resolves this by sharing common model knowledge and allowing for client-specific adaptations. Current PFL approaches for graph data primarily address heterogeneity through three main strategies during aggregation: (1) *parameter disentanglement*: splitting models into shared and personalized components (McMahan et al., 2017; Tan et al., 2023); (2) *client similarity estimation*: analyzing weights or gradients to evaluate client similarities (Xie et al., 2021); and (3) *auxiliary module calculation*: incorporating additional modules to distinguish between globally beneficial and client-specific parameters (Baek et al., 2023).

Despite their effectiveness, existing methods for PFL are confined to Euclidean space, implicitly assuming a uniform flat geometry across all client data distributions. This assumption necessitates the design of intricate mechanisms to address heterogeneity. Simple parameter disentanglement often fails, while more advanced techniques, such as client similarity estimation or auxiliary module integration, achieve better performance but incur significant computational overhead. Therefore, we revisit PFL through a geometric lens using Ricci curvature (Forman, 2003; Sun et al., 2024), which characterizes the intrinsic properties of graph structures: its sign indicates hyperbolic (negative), flat (zero), or spherical (positive) geometry, while its magnitude captures how much the space curves.

---

[1] Our method is named after Edwin Abbott's book "*Flatland: A Romance of Many Dimensions*", highlighting our insights of exploring a new perspective that maps various data distributions onto different Lorentz surfaces of hyperbolic geometry.

**Observations.** Our empirical analysis across multiple real-world datasets reveals two critical observations (Figure 3 and Figure 8): (1) client graphs predominantly exhibit *negative* Ricci curvature, indicating inherent hyperbolic structure, and (2) curvature values *vary substantially* across different clients, revealing intrinsic geometric heterogeneity that extends beyond simple statistical differences. These findings suggest that the assumption of unified Euclidean geometry in existing methods is fundamentally misaligned with the true geometric nature of graph data (Nickel & Kiela, 2018; Albert & Barabási, 2002; Khrulkov et al., 2020; Tan et al., 2023), resulting in suboptimal representations and complicating heterogeneity modeling.

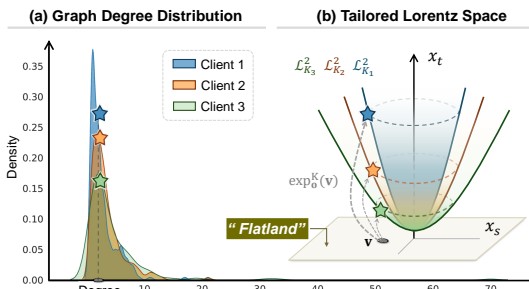

Figure 1: Example: (a) KDE of degree distributions from three CiteSeer clients (Davis et al., 2011), and (b) their respective 2D Lorentz Spaces with different curvatures $K$.

To move beyond a single Euclidean geometry, we theoretically establish the advantages of **Lorentz geometry** for PFL (Section 3.1), showing it serves as a natural testbed for two reasons: **First**, enhanced representational power: Lorentz space enables low-distortion representation of the estimated inherent graph properties (Nickel & Kiela, 2018; Peng et al., 2021; Atigh et al., 2022). When client graphs exhibit varying Ricci curvature, assigning each client an appropriate hyperbolic curvature supports a more faithful modeling (Theorem 1). See Figure 1(a), distributions are long-tailed with varying skewness. In particular, Client 1 is 'steeper' and benefits from a Lorentz space with larger curvature (smaller $K$), which offers a 'roomier' embedding environment where tail nodes can be separated. **Second**, natural heterogeneity encoding: The additional time-like dimension in Lorentz space provides a carrier to capture intrinsic geometric heterogeneity across clients (Theorem 2). In the example (Figure 1(b))[2], heterogeneity decoupling heterogeneous properties such as "*how significant is the imbalance between tail nodes and head nodes?*" can be naturally distinguished in Lorentz space through the time-like dimension $x_t$, while common information (e.g., "*the star is a tail node*") remains preserved in the space-like dimensions $\mathbf{x}_s$ (*"Flatland"*) as the same node $\mathbf{v}$.

Based on the insights, we propose FlatLand, an exploratory PFL framework that embeds client data in tailored Lorentz spaces to faithfully capture their intrinsic geometry (Section 4). Yet such embedding alone cannot resolve the heterogeneity challenge in parameter aggregation. Therefore, we further leverage the nature of the time-like dimension and develop a theoretically grounded **parameter decoupling strategy** that designates heterogeneity-related parameters as personalized, while aggregating only those carrying shared information. This design effectively mitigates heterogeneity without auxiliary modules or client-similarity estimation and preserves the validity of Lorentz geometry.

To the best of our knowledge, this is the first work to bridge hyperbolic geometry and PFL for addressing client heterogeneity in a principled manner, providing a new **succinct** and **effective** perspective that leverages geometric properties. Experimental results demonstrate that FlatLand achieves superior performance than its Euclidean counterpart, particularly in low-dimensional settings that are crucial for communication-efficient federated learning.

## 2 PRELIMINARIES

**Lorentz Manifold.** Given a $d$-dimensional Lorentz manifold $\mathcal{L}_K^d$ with a constant negative curvature $-1/K(K > 0)$, suppose a point $\mathbf{x} \in \mathcal{L}_K^d$, which has the form $\mathbf{x} = [x_t \quad \mathbf{x}_s]^\top \in \mathbb{R}^{d+1}$, where the first dimension $x_t \in \mathbb{R}$ is called *time-like* dimension and others $\mathbf{x}_s \in \mathbb{R}^d$ are *space-like* dimensions. It satisfies the following conditions: $\langle \mathbf{x}, \mathbf{x} \rangle_{\mathcal{L}} = -K$ and $x_t > 0$, where $\langle \mathbf{x}, \mathbf{y} \rangle_{\mathcal{L}} = -x_t y_t + \mathbf{x}_s^\top \mathbf{y}_s$ is the Lorentzian inner product. Note that the larger the $K$, the more the intrinsic structure of the data deviates from the flatness of Euclidean space. Formal definitions are shown in Appendix B.1.

---

[2]For convenience, all origins of Lorentz spaces in the figure are shown as the same, but in reality, their origins are not in the same location.

Typically, inputs reside in Euclidean space and need to be mapped into hyperbolic space. The way of projecting the data $\mathbf{v}^E \in \mathbb{R}^d$ in Euclidean to Lorentz space $\mathbf{x} \in \mathcal{L}_K^d$ can be simplified as [3]

$$\mathbf{x}^K = \exp_{\mathbf{o}}^K\left(\mathbf{v}^E\right) = \exp_{\mathbf{o}}^K\left([0, \mathbf{v}^E]\right) = \left(\underbrace{\cosh\left(\frac{\|\mathbf{v}^E\|_2}{\sqrt{K}}\right)}_{\text{time-like dimension } x_t}, \underbrace{\sqrt{K}\sinh\left(\frac{\|\mathbf{v}^E\|_2}{\sqrt{K}}\right)\frac{\mathbf{v}^E}{\|\mathbf{v}^E\|_2}}_{\text{space-like dimensions } \mathbf{x}_s}\right). \quad (1)$$

**Fully Lorentz Neural Networks.** Fully Lorentz network (Chen et al., 2021) has been shown to be ideal for PFL due to their reduced need for space projections, enhancing computational efficiency. These networks also incorporate Lorentz transformations (boosts and rotations), improving data heterogeneity handling and parameter interpretability (Appendix B.3).

Given an input vector $\mathbf{x} \in \mathcal{L}_K^n$, and a linear layer matrix $\mathbf{W} \in \mathbb{R}^{m \times (n+1)}$ to optimize, the fully Lorentz linear layer can be denoted as LT in a general form as $\text{LT}(\mathbf{x}; f; \mathbf{W}) := \left(\sqrt{\|f(\mathbf{Wx})\|^2 + K}, f(\mathbf{Wx})\right)^T$, where $f$ is a function like activation, dropout, and bias.

**Problem Statement.** Given clients $\mathcal{C} = \{1, 2, \ldots, C\}$, each with a dataset $\mathcal{D}_c = (\mathbf{x}_i^c, y_i^c)_{i=1}^{N_c}$ and distribution $p_c(\mathbf{x}, y)$, Personalized Federated Learning (PFL) encounters heterogeneity if $p_i(\mathbf{x}, y) \neq p_j(\mathbf{x}, y)$ for any client pair $i \neq j$, which degrades performance. In PFL, the goal is to optimize personalized models $f_c(\cdot; \boldsymbol{\theta}_c, \boldsymbol{\theta}_s)$ for each client using specific and shared parameters $\boldsymbol{\theta}_c, \boldsymbol{\theta}_s$:

$$\min_{\boldsymbol{\theta}_c|_{c=1}^C, \boldsymbol{\theta}_s} \sum_{c=1}^C \mathbb{E}_{(\mathbf{x}, y) \sim p_c(\mathbf{x}, y)}[\mathcal{L}_c(f(\mathbf{x}; \boldsymbol{\theta}_c, \boldsymbol{\theta}_s), y)] + \lambda \Omega(\boldsymbol{\theta}_c|_{c=1}^C, \boldsymbol{\theta}_s). \quad (2)$$

This function merges local loss $\mathcal{L}_c$ with regularization $\Omega$, balanced by hyperparameter $\lambda$.

---

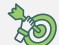 **Our goals** are

(1) to *effectively* represent the inherent properties of each local client data;

(2) to *succinctly* reflect heterogeneity among client data and facilitate the communication of shared information without requiring additional computations.

---

## 3 MOTIVATION AND INSIGHTS

We investigate PFL for graph data from a geometric perspective via Ricci curvature (Appendix B.2), which distinguishes hyperbolic (negative), flat (zero), and spherical (positive) geometries and quantifies curvature strength through its magnitude. This provides a principled measure of the intrinsic geometry of client graphs, enabling us to analyze structural differences beyond conventional statistical heterogeneity. Figure 8 and Appendix C.1 shows the results on real-world datasets, which yield **two patterns**: (1) client graphs mostly have *negative* curvature, evidencing hyperbolic structure, and (2) curvature *values vary considerably* across clients, indicating geometric heterogeneity beyond statistical differences. These findings confirm that assuming a unified Euclidean geometry leads to distorted representations and ineffective heterogeneity modeling (Nickel & Kiela, 2018; Peng et al., 2021; Atigh et al., 2022), underscoring the necessity of exploring solutions beyond Euclidean space.

### 3.1 MOTIVATION: WHY LORENTZ SPACE FOR PFL?

In this section, we bridge non-Euclidean geometry and PFL, and theoretically claim that the Lorentz geometry of hyperbolic space is particularly suitable, as it faithfully captures the non-Euclidean properties of client data and aligns with the goals outlined in Section 2.

---

[3]For clarity, all Lorentz space embeddings are denoted by $\cdot^H$. Specifically, if the curvature of the space is known as $K$, it is denoted by $\cdot^K$. In contrast, Euclidean space embeddings are denoted by $\cdot^E$. All vectors $\mathbf{x}$, if not superscripted, are assumed to be in Lorentz space.

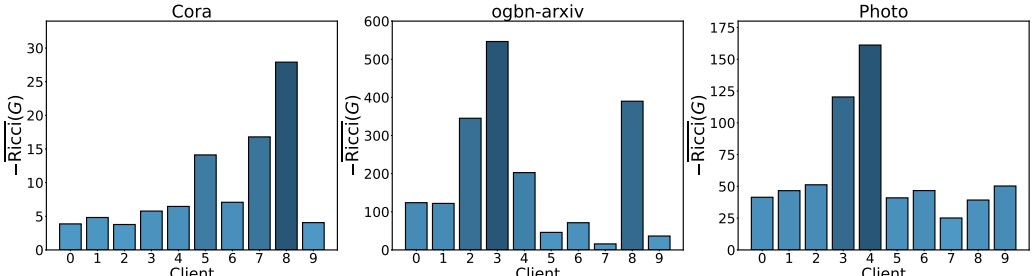

Figure 2: Averaged Forman-Ricci curvature across datasets (Cora, ogbn-arxiv, and Amazon-Photo). Higher bars indicate more pronounced non-Euclidean characteristics in these datasets.

**For Goal (1).** Prevalent non-Euclidean heterogeneity can be captured by hyperbolic curvature.

The observed *negative* curvature shows that hyperbolic space naturally fits client graphs with such properties (Yang et al., 2022), and its curvature can be adjusted to accommodate *varied* client distributions (Krioukov et al., 2010). Next, we theorize the use of hyperbolic geometry in PFL.

**Theorem 1** (Necessity of tailored curvature). *Let $\{G_c\}_{c=1}^C$ be client graphs with average Ricci curvatures $\bar{R}_c = \overline{\mathrm{Ric}}(G_c)$, and let $\mathcal{L}_K^d$ denote the $d$-dimensional hyperbolic space of constant curvature $-K < 0$. For each client $c$, let $\varepsilon_c^*(K)$ be the minimal edge distortion of any $(1+\varepsilon)$-bi-Lipschitz embedding $f_c : G_c \to L_K^d$. Then the following holds:*

$$\max_{1 \leq c \leq C} \varepsilon_c^*(K) \;\geq\; \frac{c_d}{2} \max_{1 \leq i < j \leq C} |\bar{R}_i - \bar{R}_j|, \tag{3}$$

*where $c_d > 0$ is a dimension-dependent constant (Proof in Appendix D.1).*

> **Remark 1.** *Theorem 1 indicates that if client graphs have different average Ricci curvatures, then no single hyperbolic curvature $K$ can yield simultaneously small distortion for all clients; each client requires its own tailored curvature.*

**For Goal (2).** Strong correlation between heterogeneity and hyperbolic time-like dimension.

**Theorem 2.** *Let $C \in \{1, \dots, m\}$, each client $c$ have curvature $K_c > 0$ with $\mathrm{Var}(K_C) > 0$, and $\mathbf{x} = [x_t \; \mathbf{x}_s]^\top \in \mathbb{L}_{K_c}^d$ admit hyperbolic polar coordinates $(\rho, \mathbf{u})$ with $x_t = \sqrt{K_c}\cosh(\rho/\sqrt{K_c})$, $\mathbf{x}_s = \sqrt{K_c}\sinh(\rho/\sqrt{K_c})\mathbf{u}$, $\mathbf{u} := \mathbf{x}_s/\|\mathbf{x}_s\|$. Then (1) $I(\mathbf{u}; C) = 0$; (2) $I(x_t; C) > 0$ if the pushforward measures of $T_K(\rho) := \sqrt{K}\cosh(\rho/\sqrt{K})$ differ across $K$; (3) $I\big((x_t, \mathbf{x}_s); C \mid K_c\big) = I(x_t; C \mid K_c)$ (Proof in Appendix D.2).*

> **Remark 2.** *Theorem 2 shows that the time-like component (reflected in $\mathbf{x}_t$) retains mutual information with $C$. In other words, heterogeneity across clients is encoded along the time-like dimension, and the space-like part $\mathbf{x}_s$ carries no additional client-specific information.*

### 3.2 INSIGHTS: INTRODUCE A HIGHER DIMENSION (*time dimension*) TO *"Flatland"*

In the above case, *"Flatland"* captures the common feature of a cylinder or a sphere, while a higher dimension (the third dimension) highlights the differences between the objects. Analogous to our setting, informally speaking, by introducing an additional *time-like* dimension, we can imagine each client's data residing in a unique Lorentz space (a curved world in a higher-dimensional space), where the curvature reflects the distinct distributions (objects). *"Flatland"*, $\mathbb{R}^d$ (flat), serves as a metaphor for a platform where common information (circle) is exchanged and integrated.

> *In "Flatland", a two-dimensional flat plane, the same shapes may represent the projections of various three-dimensional objects. For instance, a circle could be the projection of either a cylinder or a sphere from a higher dimension.*

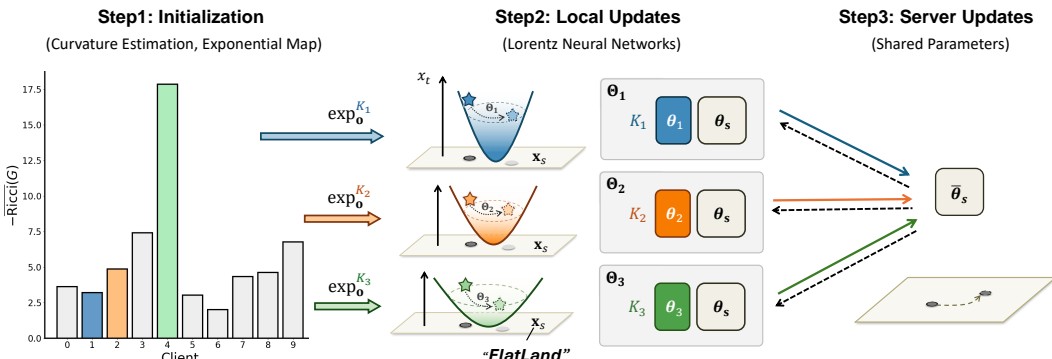

Figure 3: The FlatLand framework. It comprises three stages: (1) Initialization of curvature (Section 4.1), personalized, and shared parameters; (2) Local updates within client-specific Lorentz spaces (Appendix C.2.2); and (3) Server updates, aggregating only shared parameters while preserving personalization by keeping personalized parameters locally (Section 4).

## 4 THE FlatLand FRAMEWORK

We propose FlatLand, using tailored Lorentz spaces for each client and a well-designed parameter decoupling strategy to mitigate heterogeneity. The main steps are outlined in Figure 3 and Algorithm 2. Our method is succinct, directly built upon the basic FedAvg framework and equipped with a novel parameter decoupling strategy. It requires no additional clustering computations or auxiliary modules to address heterogeneity. Further details are provided in the Appendix C.2.

### 4.1 CURVATURE ESTIMATION

To embed the dataset $\mathcal{D}_c$ of client $c \in \mathcal{C}$ into its tailored Lorentz space $\mathcal{L}_{K_c}^d$, a suitable curvature $K_c$ should first be assigned. There are many methods that could assist in estimating the suitable curvature for various types of data (Gao et al., 2021; Ye et al., 2019; Gao et al., 2023). Given a weighted graph $G_c = (V, E, w)$ in client $c$, we adopt Forman-Ricci curvature (Appendix B.2) and the overall curvature of the graph can be calculated as follows $\overline{\mathrm{Ric}}(G) = \frac{1}{|E|} \sum_{(x,y) \in E} \mathrm{Ric}(x, y)$, where $V$ represents graph nodes and $|E|$ the number of edges, specifically, $(x, y)$ means the edge between node $x$ to node $y$. Here, we initialize $K_c$ with $\overline{\mathrm{Ric}}(G_c)$ as learnable. This procedure can be pre-calculated by each client.

### 4.2 PARAMETER DECOUPLING STRATEGY

As each client has its own Lorentz space, the curvature parameter $K$ is a personalized parameter. In this section, we focus on dividing the transformation parameters $\hat{\mathbf{M}}$ of the fully Lorentz model (Equation (2)) into personalized parameters $\boldsymbol{\theta}_c$ and shared parameters $\boldsymbol{\theta}_s$. We expect that the $\boldsymbol{\theta}_s$ transfer common information in *space-like* dimensions, while the personalized parameters $\boldsymbol{\theta}_c$ captures heterogeneous information reflected by *time-like* dimension.

First, without loss of generality, we decouple the function of the Lorentz linear layer without the functions $f$. Given input $\mathbf{x}^{(l)} = \begin{bmatrix} x_t^{(l)} & \mathbf{x}_s^{(l)} \end{bmatrix}^\top \in \mathcal{L}_K^n, x_t^{(l)} \in \mathbb{R}, \mathbf{x}_s^{(l)} \in \mathbb{R}^n$ in layer $l$. We rewrite the learnable matrix $\hat{\mathbf{M}}^{(l)}$ in Equation (2) as $\begin{bmatrix} m^{(l)} & \mathbf{M}^{(l)} \end{bmatrix} \in \mathbb{R}^{m \times (n+1)}, m^{(l)} \in \mathbb{R}^m, \mathbf{M}^{(l)} \in \mathbb{R}^{m \times n}$, the output $\mathbf{x}^{(l+1)}$ of the Lorentz linear layer could be reformulated as

$$\mathbf{x}^{(l+1)} = \mathrm{LT}(\mathbf{x}^{(l)}; \hat{\mathbf{M}}^{(l)}) = \left( \underbrace{\sqrt{\|m x_t + \mathbf{M} \mathbf{x}_s\|^2 + K}}_{\substack{\text{time-like dimension} \\ x_t^{(l+1)}}}, \underbrace{m x_t + \mathbf{M} \mathbf{x}_s}_{\substack{\text{space-like dimensions} \\ \mathbf{x}_s^{(l+1)}}} \right)^T. \qquad (4)$$

Then, we decouple the parameters as follows under the deviation from Appendix C.3. The deviation theoretically shows that, from a gradient perspective, our proposed decoupling strategy effectively separates time-like space parameters, treating them as personalized parameters.

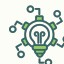 For a model $\mathcal{M}$ with $L$ layers:

$$\boldsymbol{\theta}_c = \bigcup_{l=1}^{L} \{m^{(l)}\}, \qquad \boldsymbol{\theta}_s = \bigcup_{l=1}^{L} \{M^{(l)}\},$$

where $\boldsymbol{\theta}_c$ and $\boldsymbol{\theta}_s$ denote the **personalized** and **shared** parameter sets, respectively.

## 5 ANALYSIS

In this section, we provide further analysis to substantiate our proposed method. We address three aspects: (1) **Correctness**, ensuring that client representations remain valid in Lorentz space during federated communication; (2) **Convergence**, discussing the convergence behavior of our method compared to FedAvg; (3) **Efficiency**, analyzing the computational overhead compared to the simplest FedAvg. These results highlight the good properties of our framework.

**Correctness.** Although a fully Lorentz neural network ensures that representations remain in hyperbolic space during local training Lemma 8, we further need to verify that our proposed decoupling strategy also preserves this property after server-side aggregation.

**Proposition 1.** *Let* $\hat{\mathbf{M}} = \begin{bmatrix} v & \mathbf{v}^T \\ m & \mathbf{M} \end{bmatrix}$*, where* $\hat{\mathbf{M}} \in \mathbb{R}^{(m+1)\times(n+1)}$ *and* $\Phi\left(\hat{\mathbf{M}}, \mathbf{N}\right) = \begin{bmatrix} v & \mathbf{v}^T \\ m & \mathbf{N} \end{bmatrix}$*, where* $\mathbf{N}$ *is the aggregated parameter after* $\sum_{i=1}^{C} \mathbf{M}_i$ *using the proposed decoupling strategy. For all* $\mathbf{x} \in \mathcal{L}_K^n$*,* $\hat{\mathbf{M}} \in \mathbb{R}^{(m+1)\times(n+1)}$*, and* $\mathbf{N} \in \mathbb{R}^{m\times n}$*, we have* $\mathrm{LT}\left(\mathbf{x}; \Phi\left(\hat{\mathbf{M}}, \mathbf{N}\right)\right) \in \mathcal{L}_K^m$*.*.

Proposition 1 (refer to the proof in Appendix D.3) implies that, even after the aggregation of shared parameters on the server, the transformation of any client vector $\mathbf{x} \in \mathcal{L}_K^n$ by this updated matrix will still yield results in the Lorentz space $\mathcal{L}_K^m$ with the same curvature.

**Convergence.** We analyze in Appendix D.4 whether our method hinders the convergence rate. The results show that, under the standard FedAvg scheme, our method does not affect the convergence rate, which remains at $\mathcal{O}(1/T)$.

**Efficiency.** We provide the time complexity analysis in Appendix C.4. FlatLand introduces minimal operations, like the $O(1)$ exponential map and curvature estimation, which can be mitigated by pre-computation. These minimal costs are offset by reduced communication overhead and enhanced representation in Lorentz space, making FlatLand efficient for PFL.

In summary, FlatLand preserves correctness by keeping representations in Lorentz space after aggregation, achieves the same convergence rate as FedAvg, and incurs only minimal overhead comparable to FedAvg. Besides, we further justify the rationale of our method from the perspective of Lorentz transformations in Appendix D.5.

## 6 EXPERIMENTS

In this section, we validate the effectiveness of FlatLand through experiments on *node classification* and *graph classification* on a series of benchmark datasets. The experiments are designed to address the following research questions. **RQ1.** Can FlatLand outperform personalized and hyperbolic FL baselines? **RQ2.** Can FlatLand still perform well in low-dimensional settings? **RQ3.** Can FlatLand maintain high performance under partial client participation in FL? **RQ4.** Are the proposed novel components really beneficial?

Table 1: Comparison of node classification performance across real-world datasets with varying numbers of clients. The results, presented as mean and standard deviation, are based on five separate trials. Performances that are statistically significant ($p < 0.05$) are highlighted in bold.

| | Cora | | CiteSeer | | ogbn-arxiv | | Photo | |
|---|---|---|---|---|---|---|---|---|
| # clients | 10 | 20 | 10 | 20 | 10 | 20 | 10 | 20 |
| Local ($E$) | $79.94 \pm 0.24$ | $80.30 \pm 0.25$ | $67.82 \pm 0.13$ | $65.98 \pm 0.17$ | $64.92 \pm 0.09$ | $65.06 \pm 0.05$ | $91.80 \pm 0.02$ | $90.47 \pm 0.15$ |
| Local ($L$) | $78.35 \pm 0.05$ | $80.46 \pm 0.18$ | $72.30 \pm 0.04$ | $69.52 \pm 0.25$ | $65.85 \pm 0.09$ | $66.75 \pm 0.05$ | $91.76 \pm 0.10$ | $90.12 \pm 0.20$ |
| FedAvg | $69.19 \pm 0.67$ | $69.50 \pm 3.58$ | $63.61 \pm 3.59$ | $64.68 \pm 1.83$ | $64.44 \pm 0.10$ | $63.24 \pm 0.13$ | $83.15 \pm 3.71$ | $81.35 \pm 1.04$ |
| FedProx | $60.18 \pm 7.04$ | $48.22 \pm 6.81$ | $63.33 \pm 3.25$ | $64.85 \pm 1.35$ | $64.37 \pm 0.18$ | $63.03 \pm 0.04$ | $80.92 \pm 4.64$ | $82.32 \pm 0.29$ |
| FedPer | $79.35 \pm 0.04$ | $78.01 \pm 0.32$ | $70.53 \pm 0.28$ | $66.64 \pm 0.27$ | $64.99 \pm 0.18$ | $64.66 \pm 0.11$ | $91.76 \pm 0.23$ | $90.59 \pm 0.06$ |
| GCFL | $78.66 \pm 0.27$ | $79.21 \pm 0.70$ | $69.01 \pm 0.12$ | $66.33 \pm 0.05$ | $65.09 \pm 0.08$ | $65.08 \pm 0.04$ | $92.06 \pm 0.25$ | $90.79 \pm 0.17$ |
| FedGNN | $70.12 \pm 0.99$ | $70.10 \pm 3.52$ | $55.52 \pm 3.17$ | $52.23 \pm 6.00$ | $64.21 \pm 0.32$ | $63.80 \pm 0.05$ | $87.12 \pm 2.01$ | $81.00 \pm 4.48$ |
| FedSage+ | $69.05 \pm 1.59$ | $57.97 \pm 12.6$ | $65.63 \pm 3.10$ | $65.46 \pm 0.74$ | $64.52 \pm 0.14$ | $63.31 \pm 0.20$ | $76.81 \pm 8.24$ | $80.58 \pm 1.15$ |
| FED-PUB | $\mathbf{81.54 \pm 0.12}$ | $81.75 \pm 0.56$ | $72.35 \pm 0.53$ | $67.62 \pm 0.12$ | $66.58 \pm 0.08$ | $66.64 \pm 0.12$ | $92.73 \pm 0.18$ | $91.92 \pm 0.12$ |
| FedGTA | $80.59 \pm 0.38$ | $79.01 \pm 0.31$ | $71.57 \pm 0.34$ | $69.94 \pm 0.14$ | $60.22 \pm 0.09$ | $58.74 \pm 0.14$ | $\mathbf{93.50 \pm 0.21}$ | $\mathbf{92.61 \pm 0.15}$ |
| AdaFGL | $80.09 \pm 0.00$ | $79.74 \pm 0.05$ | $72.34 \pm 0.00$ | $70.95 \pm 0.45$ | $51.77 \pm 0.36$ | $50.94 \pm 0.08$ | $89.85 \pm 0.83$ | $88.11 \pm 0.05$ |
| FedHGCN | $72.09 \pm 0.16$ | $74.67 \pm 1.50$ | $66.98 \pm 0.56$ | $64.28 \pm 0.62$ | OOM | OOM | $79.26 \pm 0.56$ | $79.57 \pm 0.10$ |
| **FlatLand (Ours)** | $80.46 \pm 0.28$ | $\mathbf{82.49 \pm 0.25}$ | $\mathbf{73.90 \pm 0.23}$ | $\mathbf{72.24 \pm 0.24}$ | $\mathbf{67.52 \pm 0.16}$ | $\mathbf{67.64 \pm 0.04}$ | $92.49 \pm 0.19$ | $91.06 \pm 0.15$ |

Table 2: Comparison of node classification performance across heterophilic datasets with varying numbers of clients. The results, presented as mean and standard deviation, are based on five separate trials. Performances that are statistically significant ($p < 0.05$) are highlighted in bold.

| | Roman-empire | | Minesweeper | | Tolokers | | Questions | |
|---|---|---|---|---|---|---|---|---|
| # clients | 10 | 20 | 10 | 20 | 10 | 20 | 10 | 20 |
| Local ($E$) | $23.54 \pm 0.26$ | $23.70 \pm 0.32$ | $70.14 \pm 0.18$ | $66.78 \pm 0.11$ | $68.86 \pm 0.26$ | $62.764 \pm 0.69$ | $59.61 \pm 0.10$ | $60.40 \pm 0.21$ |
| Local ($L$) | $54.55 \pm 0.24$ | $49.54 \pm 0.35$ | $75.02 \pm 0.21$ | $70.71 \pm 0.14$ | $72.054 \pm 0.28$ | $70.35 \pm 0.70$ | $64.47 \pm 0.10$ | $62.68 \pm 0.21$ |
| FedAvg | $35.43 \pm 0.32$ | $32.00 \pm 0.39$ | $71.18 \pm 0.02$ | $72.37 \pm 0.16$ | $54.73 \pm 0.50$ | $56.36 \pm 0.39$ | $58.91 \pm 0.22$ | $60.33 \pm 0.15$ |
| FedProx | $26.43 \pm 1.41$ | $23.12 \pm 0.49$ | $70.66 \pm 0.20$ | $71.50 \pm 0.37$ | $41.15 \pm 0.22$ | $40.42 \pm 0.62$ | $45.46 \pm 0.34$ | $46.83 \pm 0.11$ |
| FedPer | $15.51 \pm 1.13$ | $15.45 \pm 2.76$ | $65.35 \pm 7.02$ | $53.80 \pm 11.40$ | $54.97 \pm 13.23$ | $44.82 \pm 11.61$ | $59.40 \pm 9.71$ | $62.32 \pm 1.56$ |
| GCFL | $29.44 \pm 0.49$ | $26.73 \pm 0.19$ | $71.14 \pm 0.09$ | $47.77 \pm 0.14$ | $19.81 \pm 0.57$ | $17.53 \pm 0.04$ | $45.71 \pm 0.25$ | $47.47 \pm 0.21$ |
| FedGNN | $29.09 \pm 0.01$ | $26.60 \pm 0.02$ | $71.12 \pm 0.09$ | $71.71 \pm 0.27$ | $41.57 \pm 0.07$ | $40.70 \pm 0.74$ | $45.73 \pm 0.26$ | $47.46 \pm 0.25$ |
| FedSage+ | $49.07 \pm 0.00$ | $38.36 \pm 0.00$ | $72.80 \pm 0.00$ | $69.70 \pm 0.00$ | $71.31 \pm 0.00$ | $69.73 \pm 0.00$ | $65.06 \pm 0.00$ | $59.33 \pm 0.00$ |
| FED-PUB | $36.77 \pm 0.30$ | $32.67 \pm 0.39$ | $71.56 \pm 0.05$ | $70.72 \pm 0.40$ | $\mathbf{72.46 \pm 0.68}$ | $65.26 \pm 0.59$ | $54.91 \pm 0.42$ | $62.48 \pm 2.92$ |
| FedGTA | $60.94 \pm 0.19$ | $59.65 \pm 0.28$ | $64.97 \pm 0.35$ | $49.63 \pm 8.64$ | $49.97 \pm 2.68$ | $50.68 \pm 3.94$ | $53.79 \pm 0.41$ | $61.70 \pm 0.35$ |
| AdaFGL | $64.55 \pm 0.00$ | $62.42 \pm 0.26$ | $65.59 \pm 0.56$ | $51.48 \pm 7.14$ | $49.82 \pm 2.17$ | $50.62 \pm 4.19$ | $54.87 \pm 0.52$ | $62.84 \pm 0.49$ |
| FedHGCN | $55.99 \pm 0.18$ | $53.07 \pm 0.08$ | $66.37 \pm 0.05$ | $63.63 \pm 0.12$ | $65.69 \pm 0.05$ | $62.98 \pm 0.11$ | $43.21 \pm 0.08$ | $44.08 \pm 0.09$ |
| **FlatLand (Ours)** | $\mathbf{66.10 \pm 0.21}$ | $62.05 \pm 0.11$ | $\mathbf{76.34 \pm 0.05}$ | $\mathbf{74.72 \pm 0.11}$ | $71.34 \pm 0.06$ | $\mathbf{72.11 \pm 0.12}$ | $\mathbf{67.71 \pm 0.08}$ | $\mathbf{66.25 \pm 0.10}$ |

## 6.1 Experimental Setup

**Datasets and Baselines.** The details about datasets are listed in Appendix E.1. Implementation details are shown in Appendix E.2. More detailed information can be found in our anonymous repository. To assess FlatLand and demonstrate its superiority, we compare it with the following baselines: (1) Local: clients train their models locally without any communication, Local ($E$) refers to self-training in the Euclidean model, while Local ($L$) refers to training in the Lorentz model; (2) FedAvg (McMahan et al., 2017) and (3) FedProx (Li et al., 2020a): the most popular FL baselines; (4) FedPer (Arivazhagan et al., 2019): a PFL baseline with personalized model layers; (4) FedPer (Arivazhagan et al., 2019): a PFL baseline with personalized model layers; (5) GCFL (Xie et al., 2021): a PFGL baseline with client clustering and cluster-wise model aggregation; (6) FedGNN (Wu et al., 2021) and (7) FedSage+ (Zhang et al., 2021a): two FGL baselines; (8) FED-PUB (Baek et al., 2023): a PFGL baseline with personalized model aggregation and local weight masking; (9) FedGTA (Li et al., 2024b) introduces a personalized optimization strategy that leverages topology-aware local smoothing confidence and mixed neighbor features; (10) AdaFGL (Li et al., 2024a) addresses structural non-IID challenges by introducing a decoupled, two-stage personalized learning strategy ; (11) FedHGCN (Du et al., 2024): a hyperbolic FGL baseline that fails considering the heterogeneity among clients.

## 6.2 Main Experimental Results (RQ1)

**Node Classification.** We tackle node classification on *highly heterogeneous homophilic and heterophilic datasets*, with non-overlapping node partitions for each client, which most previous work fail to address. This challenge highlights our method's ability to handle heterogeneity that previous approaches could not address. Table 1 and Table 2 show that our proposed FlatLand outperforms most of the baselines with statistical significance ($p < 0.05$). (1) Local ($L$) often surpasses Local

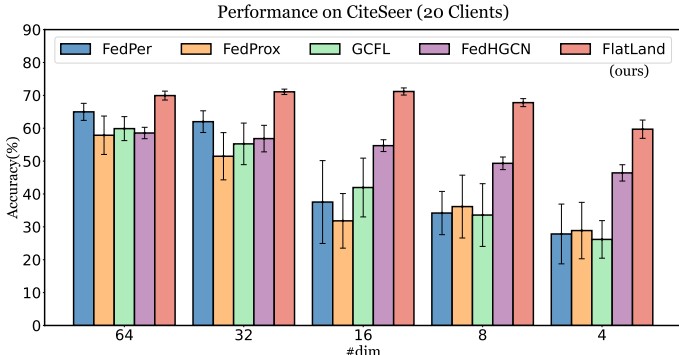
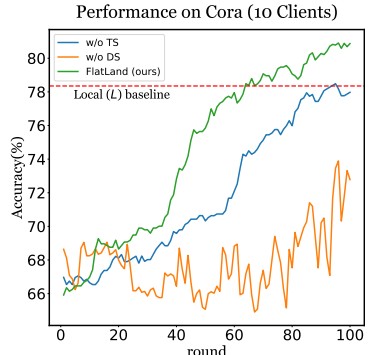

Figure 4: Performance of CiteSeer (20 clients) with varying dimensions for node classification scenario.

Figure 5: Ablation study of FlatLand on the Cora dataset.

($E$), suggesting that hyperbolic space can better represent most datasets, with the difference being particularly pronounced in heterophilic graphs. (2) Euclidean FL methods like FedAvg, FedProx and FedGNN significantly underperform self-training. Among Euclidean baselines, FED-PUB achieves the best results on homophilic graphs, while FedSage+, FedPUB, and AdaFGL perform comparably well on heterophilic graphs. 3) FedHGCN, despite operating in hyperbolic space, underperforms on heterogeneous datasets by not accounting for data heterogeneity, akin to FedAvg vs Local ($E$) in Euclidean space. Due to the quadratic time and space complexity of FedHGCN's node selection module, it can easily encounter out-of-memory (OOM) issues with large datasets like ogbn-arxiv. In contrast, our experiments show that FlatLand effectively mitigates heterogeneity and yields substantial improvements on both highly heterogeneous homophilic datasets (e.g., CiteSeer) and heterophilic datasets (e.g., Minesweeper, Roman-Empire, and Questions).

**Graph Classification.** Table 3 shows the results of the graph classification task, which is conducted with multiple datasets from one or more domains owned by different clients in each task/setting. In the single-dataset CHEM setting, Local ($L$) outperforms Local ($E$) due to inherent hyperbolic characteristics better captured by hyperbolic geometry. However, in multiple-dataset settings like BIO-CHEM-SN, Local ($L$) fails to surpass Local ($E$), potentially because not all datasets exhibit prominent hyperbolic features. With our proposed federated graph learning approach, FlatLand can significantly enhance the performance of the Lorentzian model, outperforming the Euclidean baselines, and demonstrating its effectiveness.

Table 3: Performance on graph classification tasks. The results are reported as mean $\pm$ standard deviation over five runs. Bold indicates statistical significance ($p < 0.05$).

|  | CHEM (1) | BIO-CHEM-SN (3) |
| --- | --- | --- |
| # datasets | 7 | 13 |
| Local ($E$) | $75.54 \pm 1.73$ | $67.17 \pm 1.76$ |
| Local ($L$) | $75.72 \pm 2.41$ | $65.31 \pm 2.13$ |
| FedAvg | $75.88 \pm 2.17$ | $66.91 \pm 1.94$ |
| FedProx | $76.05 \pm 1.92$ | $66.34 \pm 2.26$ |
| FedPer | $75.81 \pm 2.17$ | $66.27 \pm 2.09$ |
| GCFL | $76.49 \pm 1.23$ | $67.21 \pm 2.39$ |
| FedHGCN | $75.06 \pm 1.81$ | OOM |
| **FlatLand (Ours)** | $\mathbf{76.55 \pm 2.28}$ | $\mathbf{67.31 \pm 2.58}$ |

### 6.3 VARYING EMBEDDING DIMENSIONS (RQ2)

Lower embedding and hidden dimensions reduce the parameter transmission cost in federated learning, as fewer parameters are communicated between the server and clients during training. Considering the representational power of hyperbolic spaces in lower dimensions (Chami et al., 2019), we reduced the embedding dimension from 64 to 4 to evaluate FlatLand's ability to mitigate data heterogeneity using compact representations. Figure 4 shows the results on CiteSeer (20 clients), with similar trends observed across datasets. Dimensionality reduction from 64 to 4 had a relatively small impact on the hyperbolic methods (FlatLand and FedHGCN) compared to their Euclidean counterparts. Notably, while FedHGCN underperformed Euclidean methods at higher dimensions, it outperformed them when the dimension was reduced to 16. FlatLand consistently outperformed all other methods in different embedding dimensions, and its performance advantage over the baselines became increasingly significant as the dimensionality was reduced.

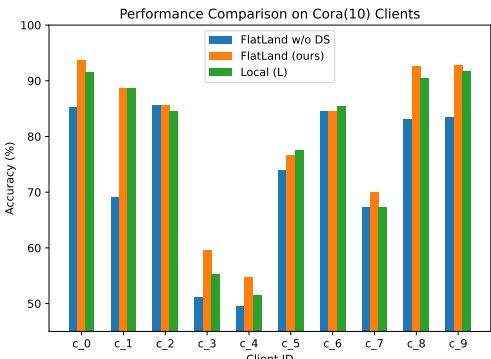 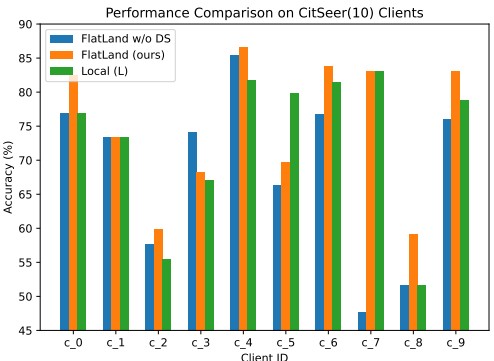

Figure 6: Performance comparison of FlatLand on Cora and CitSeer across local 10 clients.

## 6.4 ABLATION STUDY (RQ4)

To analyze the contribution of each component, we conduct ablation studies. Through ablation studies, we analyze the contribution of each component to the model's performance.

**The Benefits of Adaptive Curvature.** The "w/o TS" (without tailed space) in Figure 5 refers to setting a constant curvature of 1 for all clients instead of employing tailored curvature settings. It indicates that using a fixed hyperbolic space with constant curvature yields inferior performance compared to utilizing tailored curvatures. Furthermore, the results obtained with tailored curvatures closely approximate those of the local ($L$) setting, demonstrating the inherent effectiveness.

**The Benefits of Time-like Parameter Decoupling.** The "w/o DS" in Figure 5 refers to no parameter decoupling strategy (DS), which exhibits significant fluctuations across rounds because the aggregation process incorporates heterogeneous information, adversely affecting the results. This highlights the effectiveness of our proposed decoupling strategy and validates that the time-like dimension can effectively capture heterogeneous information. Moreover, we analyze the benefits of DS for each client's performance. As shown in Figure 6, with client IDs on the x-axis, Flatland outperforms the local method for the vast majority of clients, notably improving performance for clients with inherently poorer results, like c_8 in the CiteSeer dataset. This underscores *the necessity of federated settings for hyperbolic models*. Without our proposed DS, performance deteriorates significantly (e.g., c_7 in CiteSeer), further *validating our hypothesis that the time-like parameter encapsulates crucial heterogeneity information*.

**The Necessity of Lorentz Space.** We conducted experiments to further evaluate the necessity of using Lorentz space. Table 4 presents the results of an ablation study on the Lorentz transformation. FlatLand ($E$) represents our proposed method with a parameter decoupling strategy implemented using an Euclidean backbone. Without

Table 4: Ablation study results about the necessity of using Lorentz space to do parameter decoupling.

|  | Cora (10) | Cora (20) | CiteSeer (10) | CiteSeer (20) |
|---|---|---|---|---|
| # datasets | 10 | 20 | 10 | 20 |
| FedAvg | $69.19 \pm 0.67$ | $69.50 \pm 3.58$ | $63.61 \pm 3.59$ | $64.68 \pm 1.83$ |
| FedPer | $\underline{79.35} \pm 0.04$ | $\underline{78.01} \pm 0.32$ | $70.53 \pm 0.28$ | $\underline{66.64} \pm 0.27$ |
| FlatLand ($E$) | $78.53 \pm 0.73$ | $76.23 \pm 0.43$ | $\underline{70.68} \pm 0.52$ | $66.29 \pm 0.35$ |
| FlatLand (ours) | $\mathbf{80.46} \pm 0.28$ | $\mathbf{82.49} \pm 0.25$ | $\mathbf{73.90} \pm 0.23$ | $\mathbf{72.24} \pm 0.24$ |

Lorentz geometry, FlatLand ($E$) underperforms because the time-like parameter loses its geometric meaning. It even falls short of FedPer in most cases, which uses the classifier layer for personalization. These results validate our hypothesis and underscore the importance of hyperbolic representation.

## 7 CONCLUSIONS

In this paper, we introduce FlatLand, an exploratory PFL approach leveraging hyperbolic geometry to succinctly capture heterogeneity across clients' data distributions embedded in tailored Lorentz spaces. We propose a novel parameter decoupling strategy, which enables server-side aggregation of common information while mitigating heterogeneity interference, without client similarity estimation. This is a previously unexplored approach not only in FL but also in hyperbolic geometry.

## ETHICS STATEMENT

This work presents a novel geometric approach to personalized federated learning that enhances privacy-preserving collaborative machine learning. We identify no significant ethical concerns with our research. The proposed FlatLand framework inherently supports the ethical principles of federated learning by design: it preserves data privacy through local training without requiring raw data sharing, maintains fairness by providing personalized models that better serve diverse client populations, and promotes transparency through its theoretically grounded geometric approach. Our method does not introduce algorithmic bias beyond what may exist in the underlying data distributions, and we evaluate our approach on standard academic benchmarks without sensitive personal information. We acknowledge that federated learning systems require careful consideration of participant consent, data governance, and potential dual-use implications in deployment, though these considerations are beyond the scope of this theoretical contribution.

## REPRODUCIBILITY STATEMENT

To ensure the reproducibility of our work, we provide comprehensive details across multiple components of our submission. The theoretical foundations are fully detailed in Section 3.1 with complete mathematical formulations and proofs provided in Appendix D.2 and Appendix C.3. Our FlatLand framework implementation details are thoroughly described in Section 4, including the parameter decoupling strategy, curvature estimation procedures, and local training algorithms with pseudocode provided in Algorithm 2. All experimental configurations are specified in Section 6.2 and Appendix E.2, including hyperparameters, network architectures, training procedures, and evaluation metrics. Dataset information and preprocessing steps are detailed in Appendix E.1. We use standard benchmark datasets (Cora, CiteSeer, ogbn-arxiv, Photo for node classification; MUTAG, BZR, COX2, DHFR, PTC_MR, AIDS, NCI1, ENZYMES, DD, PROTEINS, COLLAB, IMDB-BINARY, IMDB-MULTI for graph classification) with publicly available federated learning data splits following established protocols (Baek et al., 2023). Source code implementing our method, including the Lorentz neural network layers, parameter decoupling mechanisms, and experimental scripts, is made publicly available through our anonymous repository link provided in the abstract. We provide detailed computational requirements in Appendix C.4.

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

# Contents

# A  RELATED WORK

**Personalized Federated Learning**   With statistical heterogeneity (Kairouz et al., 2021), conventional FL frameworks like FedAvg (McMahan et al., 2017) can hardly obtain a single global model that generalizes well to every client (the basic framework is shown in Appendix B.4). Motivated by this, researchers have proposed personalized FL (PFL) to train customized local models. Generally speaking, existing PFL techniques can be categorized into the following three groups: (1) techniques that personalize client models via local fine-tuning (Fallah et al., 2020; Jiang et al., 2019; Wang et al., 2019), (2) techniques that personalize client models via customized model aggregation (Huang et al., 2021; Li et al., 2021b; Luo & Wu, 2022; Sun et al., 2021; Zhang et al., 2023; 2021b), and (3) techniques that personalize client models via creating localized models/layers (Arivazhagan et al., 2019; Chen & Chao, 2022; Collins et al., 2021; Deng et al., 2020; Dinh et al., 2020; Hanzely & Richtárik, 2020; Li et al., 2021a; Mansour et al., 2020).  However, these PFL methods typically operate in Euclidean spaces to encode data samples, which can hardly capture the scale-free property and implicit hierarchical structure embedded within client data.

**Personalized Federated Graph Learning**   When applied to graph data, personalized federated graph learning (PFGL) can intuitively exhibit the problem mentioned above. For example, Xie et al. (2021) clusters clients based on gradients to aggregate models with similar data distributions. Another method (Tan et al., 2023) introduces additional personalized models to capture client-specific knowledge of graph structure. Baek et al. (2023) calculates client-client similarities to apply personalized model aggregation with local weight masking. All these methods learn node representations in Euclidean spaces, which cannot model the power-law degree distributions that widely exist in real-world graph data (Albert & Barabási, 2002; Krioukov et al., 2010). Additionally, the client clustering procedure and additional model components introduce computational overhead that may not be feasible in real-world scenarios with strict privacy constraints or limited resources.

**Hyperbolic Federated Learning**   Very few research works have considered incorporating hyperbolic spaces into federated settings. An et al. (2024) leverages hyperbolic distances to distill knowledge from the global model to the local model, to mitigate model inconsistency caused by data heterogeneity. Liao et al. (2023) applies hyperbolic prototype learning to capture the hierarchical structure among data samples. As the work most similar to our FlatLand, FedHGCN (Du et al., 2024) is a simple combination of FedAvg and hyperbolic graph neural networks along with a node selection process. Although these methods can benefit from the hyperbolic space to capture the hierarchical structure in the data, they do not have the personalization capability to adaptively model client data spaces with different curvatures. This may lead to suboptimal results when there is severe data heterogeneity. Therefore, our goal is to design a novel FL framework that can encode client data in hyperbolic spaces with adaptive curvatures using personalization techniques.

# B PRELIMINARIES

## B.1 LORENTZ MANIFOLD: FORMAL DEFINITIONS

Hyperbolic space is non-Euclidean geometry with a constant negative curvature. The curvature of hyperbolic space is a measure of how the geometry of the space deviates from the flatness of Euclidean space. The Lorentz manifold, also known as the hyperboloid model, is one of the most commonly used mathematical representations of hyperbolic space. Its greater stability for numerical optimization makes it a popular choice for hyperbolic geometry methods (Nickel & Kiela, 2018).

**Definition 1** (Lorentz Manifold). *A $d$-dimensional Lorentz manifold $\mathcal{L}_K^d$ with a negative curvature of $-1/K(K > 0)$ can be defined as the Riemannian manifold $\left(\mathbb{H}_K^d, g_\ell\right)$, where $g_\ell = \mathrm{diag}([-K, 1, \ldots, 1])$ and $\mathbb{H}_K^d = \left\{\mathbf{x} \in \mathbb{R}^{d+1} : \langle \mathbf{x}, \mathbf{x} \rangle_{\mathcal{L}} = -K, x_0 > 0\right\}$.*

**Definition 2** (Lorentzian Inner Product). *The inner product $\langle \mathbf{x}, \mathbf{y} \rangle_{\mathcal{L}}$ for $\mathbf{x}, \mathbf{y} \in \mathbb{R}^{d+1}$ can be defined as let $\langle \mathbf{x}, \mathbf{y} \rangle_{\mathcal{L}} = -x_0 y_0 + \sum_{i=1}^d x_d y_d$.*

Based on the constraint $\langle \mathbf{x}, \mathbf{x} \rangle_{\mathcal{L}} = -K$, it holds for any point $\mathbf{x} = (x_0, \mathbf{x}') \in \mathbb{R}^{d+1}$ that $\mathbf{x} \in \mathcal{L}_K^d \Leftrightarrow x_0 = \sqrt{\|\mathbf{x}'\| + K}$. The larger the value of $K$, the greater the extent to which the hyperbolic surface deviates from the Euclidean plane, as it is influenced by the larger value of $x_0$.

Next, the corresponding Lorentzian distance function for two points $\mathbf{x}, \mathbf{y} \in \mathcal{L}_K^d$ is provided as

$$d_{\mathcal{L}}^K(\mathbf{x}, \mathbf{y}) = \sqrt{K}\mathrm{arcosh}(-\langle \mathbf{x}, \mathbf{y} \rangle_{\mathcal{L}}/K). \tag{5}$$

**Definition 3** (Tangent Space). *For a point $\mathbf{x} \in \mathcal{L}_K^d$, the tangent space $\mathcal{T}_{\mathbf{x}}\mathcal{L}_K^d$ consists of all vectors orthogonal to $\mathbf{x}$, where orthogonality is defined with respect to the Lorentzian inner product( Definition 2). Hence, $\mathcal{T}_{\mathbf{x}}\mathcal{L}_K^d = \{\mathbf{v} : \langle \mathbf{x}, \mathbf{v} \rangle_{\mathcal{L}} = 0\}$.*

**Definition 4** (Exponential and Logarithmic Maps). *Let $\mathbf{v} \in \mathcal{T}_x\mathcal{L}_K^d$. The exponential map $\exp_{\mathbf{x}}^K : \mathcal{T}_{\mathbf{x}}\mathcal{L}_K^d \to \mathcal{L}_K^d$ and logarithmic map $\log_{\mathbf{x}}^K : \mathcal{L}_K^d \to \mathcal{T}_{\mathbf{x}}\mathcal{L}_K^d$ are defined as*

$$\exp_{\mathbf{x}}^K(\mathbf{v}) = \cosh\left(\frac{\|\mathbf{v}\|_{\mathcal{L}}}{\sqrt{K}}\right)\mathbf{x} + \sqrt{K}\sinh\left(\frac{\|\mathbf{v}\|_{\mathcal{L}}}{\sqrt{K}}\right)\frac{\mathbf{v}}{\|\mathbf{v}\|_{\mathcal{L}}},$$

$$\log_{\mathbf{x}}^K(\mathbf{y}) = d_{\mathcal{L}}^K(\mathbf{x}, \mathbf{y})\frac{\mathbf{y} + \frac{1}{K}\langle \mathbf{x}, \mathbf{y} \rangle_{\mathcal{L}}\mathbf{x}}{\left\|\mathbf{y} + \frac{1}{K}\langle \mathbf{x}, \mathbf{y} \rangle_{\mathcal{L}}\mathbf{x}\right\|_{\mathcal{L}}},$$

*where $\|\mathbf{v}\|_{\mathcal{L}} = \sqrt{\langle \mathbf{v}, \mathbf{v} \rangle_{\mathcal{L}}}$ denotes the norm of $\mathbf{v}$ in $\mathcal{T}_{\mathbf{x}}\mathcal{L}_K^d$.*

Particularly, for the sake of calculation, the origin of Lorentz manifold $\mathbf{o} = (\sqrt{K}, 0, 0, ..., 0) \in \mathcal{L}_K^d$ is chosen as the reference point for the exponential and logarithmic maps, which can be simplified as

$$\exp_{\mathbf{o}}^K(\mathbf{v}) = \exp_{\mathbf{o}}^K\left([0, \mathbf{v}^E]\right)$$

$$= \left(\underbrace{\cosh\left(\frac{\|\mathbf{v}^E\|_2}{\sqrt{K}}\right)}_{\text{time-like dimension}}, \underbrace{\sqrt{K}\sinh\left(\frac{\|\mathbf{v}^E\|_2}{\sqrt{K}}\right)\frac{\mathbf{v}^E}{\|\mathbf{v}^E\|_2}}_{\text{space-like dimension}}\right), \tag{6}$$

where the $(,)$ denotes concatenation and the $\cdot^E$ denotes the embedding in Euclidean space .

## B.2 FORMAN-RICCI CURVATURE

Curvature is a metric used in Riemannian geometry that expresses how far a curved line deviates from a straight line, or how much a surface deviates from planarity. In this context, knowledge of the local and global geometrical features depends on an understanding of sectional curvature and Ricci curvature, respectively (Sun et al., 2024; Ye et al., 2019).

**Sectional Curvature.** This type of curvature is determined at any given point on a manifold by examining all possible two-dimensional subspaces that intersect at that point. It provides a more

straightforward representation than the Riemann curvature tensor (Lee, 2018). Recent studies (Chen et al., 2021) often treat sectional curvature uniformly across the manifold, simplifying it to a singular constant value.

**Ricci Curvature.** Ricci curvature averages the sectional curvatures at a specific point. In graph theory, various discrete versions of Ricci curvature have been developed, such as Ollivier-Ricci curvature (Ollivier, 2009) and Forman-Ricci curvature (Forman, 2003). The Ricci curvature on graphs is intended to assess how the local structure around a graph edge deviates from that of a grid graph. Notably, the Ollivier approach provides a rougher estimate of Ricci curvature, whereas the Forman method is more combinatorial and computationally efficient.

For a weighted graph $G = (V, E, w)$, the overall Forman-Ricci curvature $\overline{\mathrm{Ric}}(G)$ can be calculated as follows:

$$\overline{\mathrm{Ric}}(G) = \frac{1}{|E|} \sum_{(i,j) \in E} \mathrm{Ric}(i, j),$$

where $|E|$ represents the cardinality of the edge set $E$ (i.e., the total number of edges), and $\mathrm{Ric}(i, j)$ is the Forman-Ricci curvature of the edge $(i, j)$, computed as (Southern et al., 2024)

$$\mathrm{Ric}(i, j) =: w_e \left( \frac{w_i}{w_e} + \frac{w_j}{w_e} - \sum_{e_l \sim i} \frac{w_i}{\sqrt{w_e w_{e_l}}} - \sum_{e_l \sim j} \frac{w_j}{\sqrt{w_e w_{e_l}}} \right)$$

where $w_e$ denotes the weight of the edge $e$, i.e, $(x, y)$, $w_i$ and $w_j$ are the weights of vertices $i$ and $j$, respectively. The sums over $e_l \sim k$ run over all edges $e_l$ incident on the vertex $k$ excluding $e$. Specifically, the curvature with vertex and edge weights set to 1 , is

$$\mathrm{Ric}(i, j) := 4 - d_i - d_j + 3|\#\Delta|,$$

where $d_i$ is the degree of node $i$ and $|\#\Delta|$ is the number of 3-cycles (i.e. triangles) containing the adjacent nodes.

Therefore, the overall Forman-Ricci curvature of the graph is the weighted average of the curvature values of all edges.

### B.3 LORENTZ TRANSFORMATIONS

In special relativity, Lorentz transformations are a family of linear transformations that describe the relationship between two coordinate frames in spacetime moving at a constant velocity relative to each other. They can be decomposed into a combination of a Lorentz Boost and a Lorentz Rotation (Moretti, 2002). The Lorentz boost, given a velocity $v \in \mathbb{R}^n$ with $\|v\| < 1$, is represented by the matrix $B$, which encodes the relative motion with constant velocity without rotation of the spatial axes. The Lorentz rotation matrix $R$ represents the rotation of spatial coordinates and is a special orthogonal matrix, i.e., $R^\top R = I$ and $\det(R) = 1$.

**Definition 5** (Lorentz Boost). *A Lorentz boost represents a change in velocity between two coordinate frames without rotation of the spatial axes. Given a velocity $\mathbf{v} \in \mathbb{R}^n$ (relative to the speed of light) with $\|\mathbf{v}\| < 1$, and the Lorentz factor $\gamma = \frac{1}{\sqrt{1 - \|\mathbf{v}\|^2}}$, the Lorentz boost matrix is defined as:*

$$\mathbf{B} = \begin{bmatrix} \gamma & -\gamma \mathbf{v}^\top \\ -\gamma \mathbf{v} & \mathbf{I} + \frac{\gamma^2}{1+\gamma} \mathbf{v}\mathbf{v}^\top \end{bmatrix} \tag{7}$$

*where $\mathbf{I}$ is the $n \times n$ identity matrix.*

A Lorentz boost describes the geometric transformation between two inertial reference frames moving at a constant relative velocity, which involves a hyperbolic rotation in the space-time plane.

**Definition 6** (Lorentz Rotation). *A Lorentz rotation describes a rotation of the spatial coordinates. The Lorentz rotation matrix is defined as:*

$$\mathbf{R} = \begin{bmatrix} 1 & \mathbf{0}^\top \\ \mathbf{0} & \tilde{\mathbf{R}} \end{bmatrix} \tag{8}$$

*where $\tilde{\mathbf{R}} \in SO(n)$ is a special orthogonal matrix satisfying $\tilde{\mathbf{R}}^\top \tilde{\mathbf{R}} = \mathbf{I}$ and $\det(\tilde{\mathbf{R}}) = 1$.*

A Lorentz rotation represents a geometric rotation or change of orientation in the spatial dimensions of the space-time manifold, while leaving the time dimension unchanged.

Both the Lorentz boost and the Lorentz rotation are linear transformations defined directly in the Lorentz model. For any point $\mathbf{x} \in \mathcal{L}_K^n$, we have $\mathbf{B}\mathbf{x} \in \mathcal{L}_K^n$ and $\mathbf{R}\mathbf{x} \in \mathcal{L}_K^n$.

### B.4   THE FEDAVG ALGORITHM

Federated Learning (FL) is a distributed learning approach that enables the training of machine learning models using data residing on local devices. A cornerstone algorithm within the FL paradigm is the FedAvg algorithm (McMahan et al., 2017). FedAvg is particularly effective for scenarios where data is decentralized and not identically distributed across participants.

---

**Algorithm 1:** FedAvg

---

**Input**   : Model parameters $\boldsymbol{\theta}$, learning rate $\eta$, and client dataset $\mathcal{D}_c$ for each client $c \in \mathcal{C}$
**Output** : Aggregated model parameters $\boldsymbol{\theta}$

1  Initialize model parameters $\boldsymbol{\theta}^{(0)}$;
2  **for** *each communication round $r$* **do**
3    **for** *each client $c$ in $\mathcal{C}$* **do**
4     Client $c$ receives global model parameters $\boldsymbol{\theta}^{(r)}$;
5     **for** *local epochs $e$* **do**
6      Compute gradients $\nabla\mathcal{L} = \nabla_{\boldsymbol{\theta}^{(r)}} \sum_{(\mathbf{x},\mathbf{y})\in\mathcal{D}_c} \mathcal{L}_c(f(\mathbf{x};\boldsymbol{\theta}^{(r)}), y)$;
7     **end**
8     Update local model $\boldsymbol{\theta}^{(r+1)} \leftarrow \boldsymbol{\theta}^{(r)} - \eta\nabla\mathcal{L}$;
9     Send $\boldsymbol{\theta}^{(r+1)}$ to the server;
10    **end**
11    $N = \sum_{c\in\mathcal{C}} |\mathcal{D}_c|$;
12    Server aggregates models $\boldsymbol{\theta}^{(r+1)} \leftarrow \frac{|\mathcal{D}_c|}{N} \sum_{c\in\mathcal{C}} \boldsymbol{\theta}_c^{(r+1)}$;
13  **end**

---

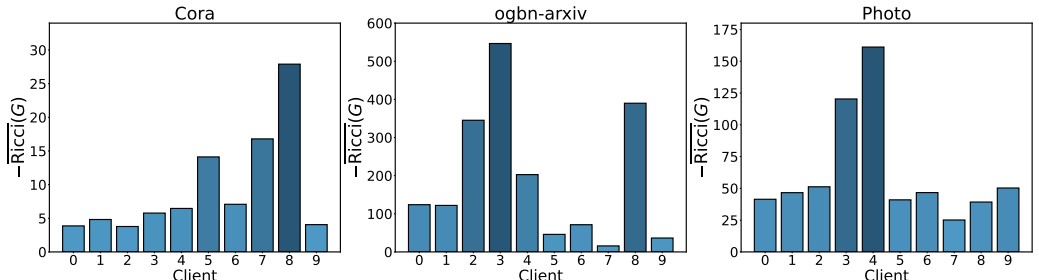

Figure 7: Averaged Forman-Ricci curvature across datasets (Cora, ogbn-arxiv, and Amazon-Photo). Higher bars indicate more pronounced non-Euclidean characteristics in these datasets.

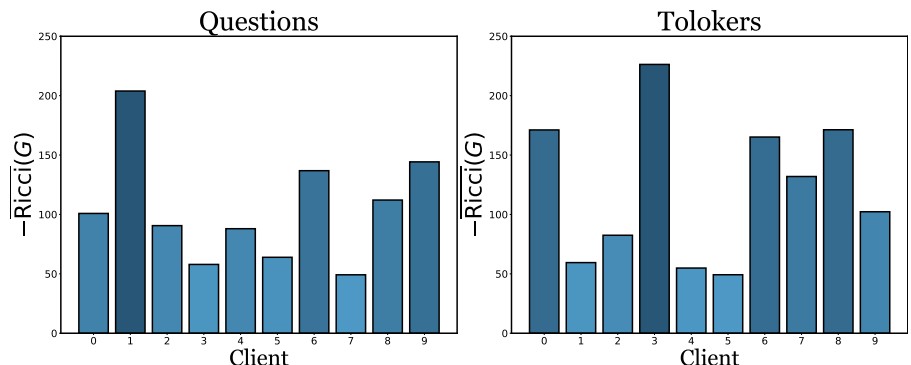

Figure 8: Averaged Forman-Ricci curvature across heterophilic datasets (Questions and Tolokers). Higher bars indicate more pronounced non-Euclidean characteristics in these datasets.

## C METHODOLOGY SUPPLEMENTARY

### C.1 STATISTICS OF FORMAN-RICCI CURVATURE IN OTHER DATASETS

We have calculated the Forman-Ricci curvature (Appendix B.2) for each client in the Cora, Photo, and ogbn-arxiv datasets, which have 10 clients each. The statistics for CiteSeer dataset are shown in Figure 3 Initialization.

### C.2 THE FlatLand ALGORITHM

This section introduces the supplementary details of our FlatLand with pseudocode shown in Algorithm 2.

#### C.2.1 OVERALL PROCESS

**S1 Initialization.** At the initial communication round $r = 0$, the parameters that need to be initialized can be divided into three groups:

(1) Curvature parameters of $C$ clients $\{K_1, K_2, ...K_C\}$ ;
(Section 4.1)

(2) Personalized parameters of $C$ clients $\{\boldsymbol{\theta}_1, \boldsymbol{\theta}_2, ..., \boldsymbol{\theta}_C\}$; (Section 4.2)

(3) Shared parameters $\overline{\boldsymbol{\theta}}_s$ of the central server.

All the parameters of client $i$ at round 0 can be written as $\boldsymbol{\Theta}_i^{(0)} = \left( K_i; \boldsymbol{\theta}_i^{(0)}; \overline{\boldsymbol{\theta}}_s^{(0)} \right)$ and server parameters as $\overline{\boldsymbol{\theta}}_s^{(0)}$.

**S2 Local updates.** Given the learning rate $\eta$ for the round $r$, each local client model performs training on the data $\mathcal{D}_i$ to minimize the task loss $\mathcal{L}(\mathcal{D}_i; \boldsymbol{\Theta}_i^{(r)})$ and then updates the parame-

ters $\boldsymbol{\Theta}_i^{(r+1)} \leftarrow \boldsymbol{\Theta}_i^{(r)} - \eta \nabla \mathcal{L}.$ (**Appendix C.2.2**)

**S3 Server updates.** After local training, only shared parameters $\boldsymbol{\theta}{s_c}^{(r+1)}$ are updated on the server for each client $c$. These are then aggregated using FedAvg: $\overline{\boldsymbol{\theta}}_s^{(r+1)} \leftarrow \frac{N_c}{N} \sum_{c=1}^{C} \boldsymbol{\theta}_{s_c}^{(r+1)}$, where $N = \sum_{c=1}^{C} N_c$. The aggregated parameters $\overline{\boldsymbol{\theta}}_s^{(r+1)}$ are subsequently distributed to the clients for the next round.

### C.2.2 LOCAL TRAINING PROCEDURE

Obtaining the curvature $K_c^{(r)}$ at round $r$, we directly project the client input $\mathbf{x}_i^E \in \mathcal{D}_c$ into its corresponding Lorentz space via the exponential map $\mathbf{x}^{K_c} = \exp_{\mathbf{o}}^{K_c}(\mathbf{x}^E)$, as shown in (Equation (1)).

Afterward, the training data are fed into the Lorentz model $\mathcal{M}$, and the output is $f((\mathbf{x}^{K_c}; \boldsymbol{\theta}_c, \boldsymbol{\theta}_s), y)$. At client $c$, the objective function is $\min_{\boldsymbol{\theta}_c|_{c=1}^C, \boldsymbol{\theta}_s} \mathcal{L}_c(f(\mathbf{x}^{K_c}; \boldsymbol{\theta}_c, \boldsymbol{\theta}_s), y) + \lambda \|\boldsymbol{\theta}_{s_c} - \overline{\boldsymbol{\theta}}_s\|_2^2$, where $\lambda$ is a hyperparameter, $\|\boldsymbol{\theta}_{s_c} - \overline{\boldsymbol{\theta}}_s\|_2^2$ is the regularized term that prevent the locally updated model $\boldsymbol{\theta}_{s_c}$ deviates too far from the server-shared parameters $\overline{\boldsymbol{\theta}}_s$.

---

**Algorithm 2:** FlatLand

**Input** : Personalized parameters $\boldsymbol{\theta}_c^{(0)}, K_c^{(0)}$ and dataset $\mathcal{D}_c$, for each client $c \in \mathcal{C}$
           Shared parameters $\overline{\boldsymbol{\theta}}_s^{(0)}$
           Learning rate $\eta$
**Output** : Client model parameters $\boldsymbol{\Theta}_c = \left(K_c; \boldsymbol{\theta}_c; \overline{\boldsymbol{\theta}}_s\right)$, for each client $c \in \mathcal{C}$
           Shared parameters $\overline{\boldsymbol{\theta}}_s$

1 Initialize model parameters: $\overline{\boldsymbol{\theta}}_s^{(0)}$ and $\boldsymbol{\Theta}_c^{(0)} = \left(K_c^{(0)}; \boldsymbol{\theta}_c^{(0)}; \overline{\boldsymbol{\theta}}_s^{(0)}\right)$, for $c \in \mathcal{C}$;
2 **for** *each communication round $r$* **do**
3     **for** *each client $c$ in $C$* **do**
4        $\mathbf{x} = \exp_{\mathbf{o}}^{K_c^{(r)}}(\mathbf{x})$, for $\mathbf{x} \in \mathcal{D}_c$;
5        Client $c$ receives global model parameters $\overline{\boldsymbol{\theta}}_s^{(r)}$;
6        $\boldsymbol{\Theta}_c^{(r)} = \left(K_c^{(r)}; \boldsymbol{\theta}_c^{(r)}; \overline{\boldsymbol{\theta}}_s^{(r)}\right)$;
7        **for** *local epochs $e$* **do**
8           Compute gradients $\nabla \mathcal{L} = \nabla_{\boldsymbol{\Theta}_c^{(r)}} \sum_{(\mathbf{x},\mathbf{y}) \in \mathcal{D}_c} \mathcal{L}_c(f(\mathbf{x}; \boldsymbol{\Theta}_c^{(r)}), y)$;
9        **end**
10        Update local model $\boldsymbol{\Theta}_c^{(r+1)} \leftarrow \boldsymbol{\Theta}_c^{(r)} - \eta \nabla \mathcal{L}$;
11        Send $\boldsymbol{\theta}_s^{(r+1)} \in \boldsymbol{\Theta}_c^{(r+1)}$ to the server;
12     **end**
13     $N = \sum_{c \in \mathcal{C}} |\mathcal{D}_c|$;
14     Server aggregates models $\overline{\boldsymbol{\theta}}_s^{(r+1)} \leftarrow \sum_{c \in \mathcal{C}} \frac{|\mathcal{D}_c|}{N} \boldsymbol{\theta}_{s_c}^{(r+1)}$;
15 **end**

---

### C.3 DERIVATION OF PARAMETERS DISENTANGLEMENT

The reformulated Lorentz neural network in layer $l$ is shown as

$$\mathbf{x}^{(l+1)} = \text{LT}(\mathbf{x}^{(l)}; \hat{\mathbf{M}}^{(l)}) \tag{9}$$

$$= \left( \underbrace{\sqrt{\|mx_t + \mathbf{M}\mathbf{x}_s\|^2 + K}}_{\substack{\text{time-like dimension} \\ x_t^{(l+1)}}}, \underbrace{mx_t + \mathbf{M}\mathbf{x}_s}_{\substack{\text{space-like dimensions} \\ \mathbf{x}_s^{(l+1)}}} \right)^T. \tag{10}$$

The loss $\mathcal{L}_c(f(\mathbf{x}; \boldsymbol{\theta}_c, \boldsymbol{\theta}_s), y)$ of client $c$, the partial derivatives can be calculated as follows:

TIME-LIKE DIMENSION $x_t^{(l+1)}$

First, we compute the partial derivative of $x_t^{(l+1)}$ with respect to the matrix $\mathbf{M}^{(l)}$ and $m^{(l)}$. Using the chain rule:

$$\frac{\partial x_t^{(l+1)}}{\partial \mathbf{M}^{(l)}} = \frac{\partial}{\partial \mathbf{M}} \sqrt{\|m^{(l)} x_t^{(l)} + \mathbf{M}^{(l)} \mathbf{x}_s^{(l)}\|^2 + K};$$

$$\frac{\partial x_t^{(l+1)}}{\partial m^{(l)}} = \frac{\partial}{\partial m} \sqrt{\|m^{(l)} x_t^{(l)} + \mathbf{M}^{(l)} \mathbf{x}_s^{(l)}\|^2 + K}.$$

Applying the chain rule, we get:

$$\begin{aligned}
\frac{\partial x_t^{(l+1)}}{\partial \mathbf{M}^{(l)}} &= \frac{1}{2} \left( \|m^{(l)} x_t^{(l)} + \mathbf{M}^{(l)} \mathbf{x}_s^{(l)}\|^2 + K \right)^{-\frac{1}{2}} \\
&\quad \cdot 2(m^{(l)} x_t^{(l)} + \mathbf{M}^{(l)} \mathbf{x}_s^{(l)}) \cdot \frac{\partial (\mathbf{M}^{(l)} \mathbf{x}_s^{(l)})}{\partial \mathbf{M}^{(l)}} \\
&= \frac{m^{(l)} x_t^{(l)} + \mathbf{M}^{(l)} \mathbf{x}_s^{(l)}}{\sqrt{\|m^{(l)} x_t^{(l)} + \mathbf{M}^{(l)} \mathbf{x}_s^{(l)}\|^2 + K}} \\
&\quad \cdot \frac{\partial (\mathbf{M}^{(l)} \mathbf{x}_s^{(l)})}{\partial \mathbf{M}^{(l)}}
\end{aligned} \tag{11}$$

$$\begin{aligned}
\frac{\partial x_t^{(l+1)}}{\partial m^{(l)}} &= \frac{1}{2} \left( \|m^{(l)} x_t^{(l)} + \mathbf{M}^{(l)} \mathbf{x}_s^{(l)}\|^2 + K \right)^{-\frac{1}{2}} \\
&\quad \cdot 2(m^{(l)} x_t^{(l)} + \mathbf{M}^{(l)} \mathbf{x}_s^{(l)}) \cdot \frac{\partial (m^{(l)} \mathbf{x}_t^{(l)})}{\partial \mathbf{M}^{(l)}} \\
&= \frac{(m^{(l)} x_t^{(l)} + \mathbf{M}^{(l)} \mathbf{x}_s^{(l)})}{\sqrt{\|m^{(l)} x_t^{(l)} + \mathbf{M}^{(l)} \mathbf{x}_s^{(l)}\|^2 + K}} \cdot x_t^{(l)}
\end{aligned} \tag{12}$$

SPACE-LIKE DIMENSION $\mathbf{x}_s^{(l+1)}$

Assume that the update rule for the space-like vector $\mathbf{x}_s^{(l+1)}$ is given by the following formula:

$$\mathbf{x}_s^{(l+1)} = m^{(l)} x_t^{(l)} + \mathbf{M}^{(l)} \mathbf{x}_s^{(l)}$$

Similarly, we have

$$\frac{\partial \mathbf{x}_s^{(l+1)}}{\partial \mathbf{M}^{(l)}} = \frac{\partial \left( \mathbf{M}^{(l)} \mathbf{x}_s^{(l)} \right)}{\partial \mathbf{M}^{(l)}}, \quad \frac{\partial \mathbf{x}_s^{(l+1)}}{\partial m^{(l)}} = \frac{\partial \left( m^{(l)} \mathbf{x}_t^{(l)} \right)}{\partial m^{(l)}}. \tag{13}$$

*"Flatland"* is the space of dimension $1 : n$, serving as a metaphor for a platform where common information is exchanged and integrated. The same space-like dimension transformation $\mathbf{x}_s^{(l)} \to \mathbf{x}_s^{(l+1)}$, i.e., $\mathbf{x}_s^{(l)} \to \left( \mathbf{M}^{(l)} \mathbf{x}_s^{(l)} + m^{(l)} \mathbf{x}_s^{(l)} \right)$ in different client with different curvatures, it is easy to know that the gradient of the parameter $m$ is only related to $x_t$.

For better illustration, here, we let $\mathbf{x}^{(l)} \in \mathcal{L}_K^n$, $\mathbf{x}^{(l+1)} \in \mathcal{L}_K^n$, and $\hat{\mathbf{M}}^{(l)} \in \mathbb{R}^{(n+1) \times (n+1)}$. The introduced *"Flatland"* $\mathbb{R}^n$ is defined as a manifold spanning dimensions $1$ to $n$. This construct serves as a metaphorical platform for the exchange and integration of common information, and $x_t$ serves as the heterogeneous information. Consider the same transformation of a space-like vector $\mathbf{x}_s^{(l)}$ to $\mathbf{x}_s^{(l+1)}$ in different clients, formulated as

$$\mathbf{x}_s^{(l)} \to \left( \mathbf{M}^{(l)} \mathbf{x}_s^{(l)} + m^{(l)} \mathbf{x}_s^{(l)} \right),$$

**Remark 3.** *Note that the key component capturing data heterogeneity in layer $l$ is the time-like dimension of the hyperbolic embedding $x_t^{(l)}$, not original input $x_t^{(0)}$. After each layer of the Lorentz neural network, the input vectors are transformed into a new hyperbolic embedding, with the time-like dimension updated as $x_t^{(l+1)} = \sqrt{||mx_t^{(l)} + \mathbf{M}\mathbf{x}_s^{(l)}||^2 + K}$. This means that the component at layer $(l+1)$ that carries the heterogeneity information is transferred as $x_t^{(l+1)}$. Thus, the parameters associated with $x_t^{(l+1)}$ are naturally aligned with the heterogeneity information. This ensures that our decoupling strategy is consistent with the theoretical derivation and can be effectively applied across multiple layers.*

## C.4 Time and Space Complexity Compared with FedAvg

We analyze the computational complexity of FlatLand compared to FedAvg, which gives insight into the scalability.

**Local Update.** The additional operations in FlatLand's local update phase compared with FedAvg - curvature estimation (Section 4.1), exponential map (line 4 in Algorithm 2, Equation 6). Notably, the curvature estimation can be *pre-computed* since each client's data distribution corresponds to a constant curvature value. For exponential map, the transformation only requires *a single* non-linear mapping operation based on the norm of input samples with the time complexity of $O(1)$. These norms can also be *pre-computed and cached*. Therefore, while FlatLand introduces these additional steps compared to FedAvg, their practical computational overhead is limited due to pre-computation opportunities and constant-time operations.

**Aggregation.** FlatLand and FedAvg have the same aggregation time complexity when the hidden embedding dimension is the same. Though FlatLand introduces extra time-like space parameters, it only aggregates shared parameters $\boldsymbol{\theta}_s$ while maintaining personalized parameters. The overhead of the shared parameters is the same. Moreover, FlatLand can perform better in low dimensionality (Section 6.3), which potentially reduces practical communication costs.

**Space Requirements and Storage.** FlatLand requires extra $O(d+1)$ storage per client compared to FedAvg due to the additional time-like dimension and curvature parameter, where $d$ is the hidden dimension. Since typically $d$ is small, the increase in storage is small. Moreover, FlatLand demonstrates superior performance even in low-dimensional settings compared with the Euclidean counterparts, which further limits the practical storage overhead.

This analysis suggests that FlatLand can balance the trade-off between computational overhead and model effectiveness, showing the scalability for the increase in clients. While it introduces additional operations in local computations, these overheads are limited and offer significant optimization opportunities through pre-computation and caching strategies. The method compensates for these minimal costs through reduced communication overhead and enhanced representation capabilities in the Lorentz space, making it a practical and efficient choice for personalized federated learning applications.

## D   THEORETICAL ANALYSIS SUPPLEMENTARY

### D.1   PROOF FOR THEOREM 1

**Lemma 1.** *Let $G = (V, E)$ be a connected, simple, unweighted graph with maximum degree $\Delta < \infty$ and average Forman–Ricci curvature $\bar{R} = \overline{\mathrm{Ric}}(G)$ (edge-average). Define the averaged ball counts $V_G(r) := \frac{1}{|V|} \sum_{v \in V} |B_r^G(v)|$ for $r = 0, 1, 2$. Then*

$$\Delta^2 V_G(1) := V_G(2) - 2V_G(1) + V_G(0) \geq 1 - \frac{|E|}{|V|} \bar{R}. \tag{14}$$

*Proof.* For the unweighted, no-higher-cell case, the Forman–Ricci curvature on an edge $(u, v)$ is $\mathrm{Ric}(u, v) = 4 - \deg(u) - \deg(v)$. Counting non-backtracking two-step walks yields

$$\frac{1}{|V|} \sum_{v \in V} \sum_{u \sim v} (\deg(u) - 1) = \frac{1}{|V|} \sum_{(u,v) \in E} \big(\deg(u) + \deg(v) - 2\big) = \frac{1}{|V|} \Big(2|E| - \sum_{e \in E} \mathrm{Ric}(e)\Big).$$

Using $V_G(0) = 1$, $V_G(1) = 1 + \frac{1}{|V|} \sum_v \deg(v) = 1 + \frac{2|E|}{|V|}$, and adding the two-step term gives Equation (14). Short cycles only increase $V_G(2)$, hence the inequality. $\square$

**Lemma 2** (Gray (2003)). *Let $\mathcal{L}_K^d$ be the $d$-dimensional hyperbolic space with constant curvature $-K < 0$ and $\mathrm{Vol}_\mathcal{L}(B_\rho)$ the volume of a radius-$\rho$ ball. For small $\rho$,*

$$\mathrm{Vol}_\mathcal{L}(B_\rho) = \omega_d \, \rho^d \Big(1 + \alpha_d \, (d-1)K \, \rho^2 + O(\rho^4)\Big), \quad \alpha_d = \frac{d}{6(d+2)}. \tag{15}$$

*Taking the discrete second difference in $\rho \in \{0, 1, 2\}$ gives*

$$\Delta^2 \mathrm{Vol}_\mathcal{L}(1) = \mathrm{Vol}_\mathcal{L}(B_2) - 2\mathrm{Vol}_\mathcal{L}(B_1) + \mathrm{Vol}_\mathcal{L}(B_0) = C_d \, (d-1)K + O(1), \tag{16}$$

*where $C_d := \omega_d \, \alpha_d \, \Delta^2[\rho^{d+2}]_{\rho=1} > 0$.*

**Lemma 3** (Local bi-Lipschitz sandwich). *Let $f : V \to \mathcal{L}_K^d$ be $(1 + \varepsilon)$-bi-Lipschitz on graph balls of radius 2, i.e., $d_\mathcal{L}\big(f(x), f(y)\big) \in [(1 + \varepsilon)^{-1} d_G(x, y), (1 + \varepsilon) d_G(x, y)]$ whenever $d_G(x, y) \leq 2$. Then there exist constants $A_{d,\Delta}, B_{d,\Delta} > 0$ such that*

$$A_{d,\Delta} \, \mathrm{Vol}_\mathcal{L}\big(B_{(1-\varepsilon)\rho}\big) \leq V_G(\rho) \leq B_{d,\Delta} \, \mathrm{Vol}_\mathcal{L}\big(B_{(1+\varepsilon)\rho}\big), \qquad \rho \in \{0, 1, 2\}. \tag{17}$$

*Consequently, taking discrete second differences and Taylor-expanding at $\varepsilon = 0$,*

$$\Delta^2 V_G(1) = \Gamma_{d,\Delta} \, (d-1)K \, \pm \, \Lambda_{d,\Delta} \, \varepsilon \, + \, O(\varepsilon^2), \tag{18}$$

*for some $\Gamma_{d,\Delta}, \Lambda_{d,\Delta} > 0$.*

*Proof.* Inclusions $f(B_\rho^G(v)) \subset B_{(1+\varepsilon)\rho}^\mathcal{L}(f(v))$ and $B_{(1-\varepsilon)\rho}^\mathcal{L}(f(v)) \subset f(B_\rho^G(v))$ follow from the bi-Lipschitz bounds. Degree bound $\Delta$ gives packing/covering constants relating point counts and volumes, yielding the sandwich; apply $\Delta^2$ in $\rho$ and a first-order Taylor expansion in $\varepsilon$. $\square$

**Lemma 4** (Curvature mismatch $\Rightarrow$ local distortion). *Under the conditions of Lemma 1–Lemma 3, there exist constants $c_d > 0$ and $C_{d,\Delta} > 0$ such that any $(1 + \varepsilon)$-bi-Lipschitz $f$ on radius-2 balls satisfies*

$$\varepsilon \geq c_d \big|\bar{R} + (d-1)K\big| - C_{d,\Delta} \, \varepsilon^2. \tag{19}$$

*In particular, for $\varepsilon \in (0, \varepsilon_0(d, \Delta))$,*

$$\varepsilon \geq \tfrac{1}{2} c_d \big|\bar{R} + (d-1)K\big|. \tag{20}$$

*Proof.* Combine Equation (14) and Equation (18); rearrange to isolate $|\bar{R} + (d-1)K|$ in terms of $\varepsilon$, absorbing constants into $c_d, C_{d,\Delta}$. For sufficiently small $\varepsilon$, the quadratic term is dominated, giving Equation (20). $\square$

**Theorem 3** (Recall of Theorem 1)**.** *Let $\{G_c\}_{c=1}^C$ be client graphs with average Ricci curvatures $\bar{R}_c = \overline{\mathrm{Ric}}(G_c)$, and let $\mathcal{L}_K^d$ denote the $d$-dimensional hyperbolic space of constant curvature $-K < 0$. For each client $c$, let $\varepsilon_c^*(K)$ be the minimal edge distortion of any $(1 + \varepsilon)$-bi-Lipschitz embedding $f_c : G_c \to L_K^d$. Then the following holds:*

$$\max_{1 \leq c \leq C} \varepsilon_c^*(K) \;\geq\; \frac{c_d}{2} \max_{1 \leq i < j \leq C} |\bar{R}_i - \bar{R}_j|, \tag{21}$$

*where $c_d > 0$ is a dimension-dependent constant.*

*Proof.* For client $c$, Lemma 4 applied to $G_c$ gives (for small optimal distortion)

$$\varepsilon_c^*(K) \;\geq\; c_d \big| \bar{R}_c + (d-1)K \big|. \tag{22}$$

For any $i \neq j$ set $a_i := \bar{R}_i + (d-1)K$, $a_j := \bar{R}_j + (d-1)K$. Then $a_i - a_j = \bar{R}_i - \bar{R}_j$, so $\max\{|a_i|, |a_j|\} \geq \frac{1}{2}|a_i - a_j| = \frac{1}{2}|\bar{R}_i - \bar{R}_j|$. Taking the maximum over all pairs $(i, j)$ yields

$$\max_{1 \leq c \leq C} \big| \bar{R}_c + (d-1)K \big| \;\geq\; \frac{1}{2} \max_{1 \leq i < j \leq C} |\bar{R}_i - \bar{R}_j|. \tag{23}$$

Combining Equation (22) and Equation (23) gives

$$\max_{1 \leq c \leq C} \varepsilon_c^*(K) \;\geq\; c_d \max_c \big| \bar{R}_c + (d-1)K \big| \;\geq\; \frac{c_d}{2} \max_{1 \leq i < j \leq C} |\bar{R}_i - \bar{R}_j|,$$

which is the desired inequality. If $\{\bar{R}_c\}$ are not all equal, the right-hand side is strictly positive, hence a single $K$ cannot make all clients' distortions simultaneously small. □

### D.2 PROOF FOR THEOREM 2

For each client $c$ with curvature $K_c > 0$, consider $\mathbf{x} \in \mathbb{L}_{K_c}^d$ expressed in hyperbolic polar coordinates $(\rho, \mathbf{u})$, where $\rho \geq 0$ is the radial coordinate and $\mathbf{u} \in \Omega := \mathbb{S}^{d-1}$ is the angular coordinate (unit direction vector). We denote by $p(\rho, \mathbf{u} \mid K_c)$ the joint distribution of $(\rho, \mathbf{u})$ given $K_c$.

**Assumption 1** (Isotropy at the basepoint (Krioukov et al., 2010))**.** *For each client $c$ with curvature $K_c$, $p(\rho, \mathbf{u} \mid K_c)$ is $G_o$–isotropic at the basepoint $o$, i.e. $p(g \cdot \mathbf{x} \mid K_c) = p(\mathbf{x} \mid K_c) \quad \forall g \in G_o \simeq SO(d)$. Equivalently, the joint law factorizes as*

$$p(\rho, \mathbf{u} \mid K_c) = p_\rho(\rho \mid K_c)\, p_\Omega(\mathbf{u}),$$

*where $p_\rho(\rho \mid K_c)$ is the radial density depending on $K_c$, and $p_\Omega(\mathbf{u})$ is the $SO(d)$–invariant angular density on $\Omega$, independent of $K_c$ (in particular, uniform on $\mathbb{S}^{d-1}$).*

**Lemma 5.** *Under Assumption 1, the angular component $\mathbf{u}$ is independent of the client identity $C$. In particular, $I(\mathbf{u}; C) = 0$.*

*Proof.* Since $K_c$ is a deterministic function of $C$, for any $c$ we compute

$$
\begin{aligned}
p(\mathbf{u} \mid C = c) &= \int p(\rho, \mathbf{u} \mid C = c)\, d\rho \\
&= \int p(\rho, \mathbf{u} \mid K_c)\, d\rho \\
&= \int p_\rho(\rho \mid K_c)\, p_\Omega(\mathbf{u})\, d\rho \qquad \text{(Assumption 1)} \\
&= p_\Omega(\mathbf{u}) \int p_\rho(\rho \mid K_c)\, d\rho \\
&= p_\Omega(\mathbf{u}) \tag{24}
\end{aligned}
$$

Therefore the marginal satisfies

$$p(\mathbf{u}) = \sum_c p(C = c)\, p(\mathbf{u} \mid C = c) = \sum_c p(C = c)\, p_\Omega(\mathbf{u}) = p_\Omega(\mathbf{u}).$$

Substituting into the KL formulation of mutual information,

$$I(\mathbf{u}; C) = \sum_c p(C = c) \, D_{\mathrm{KL}}\big(p(\mathbf{u} \mid C = c) \,\|\, p(\mathbf{u})\big)$$

$$= \sum_c p(C = c) \int_\Omega p(\mathbf{u} \mid C = c) \, \log \frac{p(\mathbf{u} \mid C = c)}{p(\mathbf{u})} \, d\sigma(\mathbf{u})$$

$$= \sum_c p(C = c) \int_\Omega p_\Omega(\mathbf{u}) \, \log \frac{p_\Omega(\mathbf{u})}{p_\Omega(\mathbf{u})} \, d\sigma(\mathbf{u})$$

$$= \sum_c p(C = c) \, D_{\mathrm{KL}}(p_\Omega \,\|\, p_\Omega) = 0. \tag{25}$$

This completes the proof.

$\square$

**Lemma 6.** *Let $x_t = T_{K_c}(\rho) := \sqrt{K_c} \cosh(\rho/\sqrt{K_c})$ with $\rho \sim p_\rho(\cdot \mid K_c)$. Here $(T_{K_c})_\# p_\rho(\cdot \mid K_c)$ denotes the pushforward law of $p_\rho(\cdot \mid K_c)$ through $T_{K_c}$, i.e. the distribution of $x_t$. If there exist $c_1 \neq c_2$ with $\mathbb{P}(C = c_i) > 0$ such that $(T_{K_{c_1}})_\# p_\rho(\cdot \mid K_{c_1}) \neq (T_{K_{c_2}})_\# p_\rho(\cdot \mid K_{c_2})$, then $x_t$ is not independent of $C$ and hence $I(x_t; C) > 0$.*

*Proof.* Conditioned on $C = c$, the curvature is fixed to $K_c$, and the law of $x_t$ is the pushforward of $p_\rho(\cdot \mid K_c)$ under $T_{K_c}$: for every Borel set $A \subset \mathbb{R}$,

$$\mathbb{P}(x_t \in A \mid C = c) = \mathbb{P}\big(T_{K_c}(\rho) \in A \mid K_c\big)$$

$$= \big[(T_{K_c})_\# p_\rho(\cdot \mid K_c)\big](A). \tag{26}$$

By the assumption of differing pushforward laws, there exist $c_1 \neq c_2$ with $\Pr(C = c_i) > 0$ such that

$$p(x_t \mid C = c_1) \neq p(x_t \mid C = c_2).$$

Let $p(x_t) = \sum_c p(C = c) \, p(x_t \mid C = c)$ denote the marginal of $x_t$. Using the KL expansion of mutual information,

$$I(x_t; C) = \sum_c p(C = c) \, D_{\mathrm{KL}}\big(p(x_t \mid C = c) \,\|\, p(x_t)\big)$$

$$= \sum_c p(C = c) \int_{\mathbb{R}} p(x_t \mid C = c) \, \log \frac{p(x_t \mid C = c)}{p(x_t)} \, dx_t. \tag{27}$$

If $p(x_t \mid C = c_1) \neq p(x_t \mid C = c_2)$ and both $p(C = c_i) > 0$, then at least one of $p(x_t \mid C = c_i)$ differs from the mixture $p(x_t)$; by Gibbs' inequality, the corresponding KL term is strictly positive:

$$D_{\mathrm{KL}}\big(p(x_t \mid C = c_i) \,\|\, p(x_t)\big) > 0 \quad \text{for some } i \in \{1, 2\}.$$

Since all KL terms are nonnegative, Equation (27) yields $I(x_t; C) > 0$.

Equivalently, from Equation (26), there exists a Borel set $A$ such that $\mathbb{P}(x_t \in A \mid C = c_1) \neq \mathbb{P}(x_t \in A \mid C = c_2)$, which already rules out independence and thus forces $I(x_t; C) > 0$. This completes the proof. $\square$

**Lemma 7.** *Under Assumption 1, we have $I\big((x_t, \mathbf{x}_s); C \mid K_c\big) = I(x_t; C \mid K_c)$.*

*Proof.* By Assumption 1, $\mathbf{u}$ is independent of $(K_c, \rho)$, hence of any measurable function thereof; in particular $\mathbf{u} \perp (x_t, K_c, C)$. Using the chain rule,

$$I(\mathbf{x}_s; C \mid x_t, K_c) = I(r_s, \mathbf{u}; C \mid x_t, K_c)$$

$$= I(\mathbf{u}; C \mid x_t, K_c) + I(r_s; C \mid x_t, K_c, \mathbf{u})$$

$$= 0 + 0 \quad \text{(since } \mathbf{u} \perp (x_t, K_c, C) \text{ and } r_s \text{ is deterministic given } (x_t, K_c))$$

$$= 0.$$

Finally, apply the chain rule conditioned on $K_c$:

$$I\big((x_t, \mathbf{x}_s); C \mid K_c\big) = I(x_t; C \mid K_c) + I(\mathbf{x}_s; C \mid x_t, K_c) = I(x_t; C \mid K_c),$$

which proves the stated conclusion. $\square$

**Remark 4.** *Lemma 7 shows that, conditional on curvature $K_c$, all heterogeneity is concentrated in the time-like dimension $x_t$ ; the space-like part $\mathbf{x}_s$ carries no additional client-specific information.*

**Theorem 4** (Recall of Theorem 2). *Let $C \in \{1, \dots, m\}$, each client $c$ have curvature $K_c > 0$ with $\mathrm{Var}(K_C) > 0$, and $\mathbf{x} = [x_t \ \mathbf{x}_s]^\top \in \mathbb{L}_{K_c}^d$ admit hyperbolic polar coordinates $(\rho, \mathbf{u})$ with $x_t = \sqrt{K_c} \cosh(\rho/\sqrt{K_c})$, $\mathbf{x}_s = \sqrt{K_c} \sinh(\rho/\sqrt{K_c})\,\mathbf{u}$, $\mathbf{u} := \mathbf{x}_s/\|\mathbf{x}_s\|$. Then (1) $I(\mathbf{u}; C) = 0$; (2) $I(x_t; C) > 0$ if the pushforward measures of $T_K(\rho) := \sqrt{K} \cosh(\rho/\sqrt{K})$ differ across $K$; (3) $I\big((x_t, \mathbf{x}_s); C \mid K_c\big) = I(x_t; C \mid K_c)$.*

*Proof.* The first claim follows from Lemma 5, the second from Lemma 6, and the third from Lemma 7. □

### D.3 PROOF FOR PROPOSITION 1

**Lemma 8.** *Let $\mathcal{L}_K^n$ denote the $n$-dimensional Lorentz space with curvature $K$. For any $\mathbf{x} \in \mathcal{L}_K^n$ and any transformation matrix $\mathbf{M} \in \mathbb{R}^{(m+1)\times(n+1)}$, the Lorentz transformation LT preserves the Lorentz structure, i.e., $\mathrm{LT}(\mathbf{x}; \mathbf{M}) \in \mathcal{L}_K^m$.*

*Proof.* Let $\mathbf{x} \in \mathcal{L}_K^n$. By the definition of the Lorentz transformation LT, we compute the Lorentzian inner product:

$$\langle \mathrm{LT}(\mathbf{x}; \mathbf{M}), \mathrm{LT}(\mathbf{x}; \mathbf{M}) \rangle_{\mathcal{L}} = -K.$$

Since this condition characterizes membership in the Lorentz space $\mathcal{L}_K^m$, it follows that $\mathrm{LT}(\mathbf{x}; \mathbf{M}) \in \mathcal{L}_K^m$. This completes the proof. □

**Proposition 2** (Recall of Proposition 1). *Let $\hat{\mathbf{M}} = \begin{bmatrix} v & \mathbf{v}^T \\ m & \mathbf{M} \end{bmatrix}$, where $\hat{\mathbf{M}} \in \mathbb{R}^{(m+1)\times(n+1)}$ and $\Phi\big(\hat{\mathbf{M}}, \mathbf{N}\big) = \begin{bmatrix} v & \mathbf{v}^T \\ m & \mathbf{N} \end{bmatrix}$, where $\mathbf{N}$ is the aggregated parameter after $\sum_{i=1}^{C} \mathbf{M}_i$ using the proposed decoupling strategy. For all $\mathbf{x} \in \mathcal{L}_K^n$, $\hat{\mathbf{M}} \in \mathbb{R}^{(m+1)\times(n+1)}$, and $\mathbf{N} \in \mathbb{R}^{m \times n}$, we have $\mathrm{LT}\big(\mathbf{x}; \Phi\big(\hat{\mathbf{M}}, \mathbf{N}\big)\big) \in \mathcal{L}_K^m$.*

*Proof.* Let $\mathbf{x} = \begin{bmatrix} x_t \\ \mathbf{x}_s \end{bmatrix} \in \mathcal{L}_K^n$, where $x_t \in \mathbb{R}$, $\mathbf{x}_s \in \mathbb{R}^n$. According to Equation (4), we have:

$$\mathrm{LT}\big(\mathbf{x}; \Phi(\hat{\mathbf{M}}, \mathbf{N})\big) = \begin{bmatrix} \sqrt{\|mx_t + \mathbf{N}\mathbf{x}_s\|^2 + K} \\ mx_t + \mathbf{N}\mathbf{x}_s \end{bmatrix}$$

We need to prove that $\mathrm{LT}(\mathbf{x}; \Phi(\hat{\mathbf{M}}, \mathbf{N})) \in \mathcal{L}_K^m$, i.e., to prove that it satisfies the definition condition of the Lorentz manifold $\langle \cdot, \cdot \rangle_{\mathcal{L}} = -K$:

$$
\begin{aligned}
&\left\langle \mathrm{LT}\big(\mathbf{x}; \Phi(\hat{\mathbf{M}}, \mathbf{N})\big), \mathrm{LT}\big(\mathbf{x}; \Phi(\hat{\mathbf{M}}, \mathbf{N})\big) \right\rangle_{\mathcal{L}} \\
&= \left\langle \begin{bmatrix} \sqrt{\|mx_t + \mathbf{N}\mathbf{x}_s\|^2 + K} \\ mx_t + \mathbf{N}\mathbf{x}_s \end{bmatrix}, \begin{bmatrix} \sqrt{\|mx_t + \mathbf{N}\mathbf{x}_s\|^2 + K} \\ mx_t + \mathbf{N}\mathbf{x}_s \end{bmatrix} \right\rangle_{\mathcal{L}} \quad \text{(Definition 2)} \\
&= -\left( \sqrt{\|mx_t + \mathbf{N}\mathbf{x}_s\|^2 + K} \right)^2 + \|mx_t + \mathbf{N}\mathbf{x}_s\|^2 \\
&= -K
\end{aligned}
$$

Therefore, we have proved that $\mathrm{LT}\big(\mathbf{x}; \Phi(\hat{\mathbf{M}}, \mathbf{N})\big) \in \mathcal{L}_K^m$. □

### D.4    Convergence Analysis

FedAvg converges to the global optimum at a rate of $O(\frac{1}{T})$ for strongly convex and smooth functions and non-iid data. When the learning rate is sufficiently small, the effect of $E$ steps of local updates is similar to a step update with a larger learning rate (Li et al., 2020b).

In this section, we demonstrate that FlatLand achieves a convergence rate of $O(\frac{1}{T})$ without regularization, which is consistent with FedAvg. Furthermore, when incorporating regularization similar to FedProx (Li et al., 2020a), the convergence rate can be bounded by a constant that reflects the degree of data heterogeneity, analogous to FedProx's theoretical guarantees. This analysis confirms that our special geometric enhanced decoupling strategy maintains the overall convergence properties while addressing the challenges of heterogeneous data distribution.

To simplify the analysis, we consider each client conducts full batch gradient descent with one step. At client $c$, the objective function can be generally written as

$$\min_{\boldsymbol{\theta}_c|_{c=1}^C, \boldsymbol{\theta}_s} \mathcal{L}_c(f(\mathbf{x}^{K_c}; \boldsymbol{\theta}_c, \boldsymbol{\theta}_s), y) + \lambda\|\boldsymbol{\theta}_{s_c} - \overline{\boldsymbol{\theta}}_s\|_2^2, \tag{28}$$

where $\lambda$ is a hyperparameter, $y \in \mathcal{Y}$, $\|\boldsymbol{\theta}_{s_c} - \overline{\boldsymbol{\theta}}_s\|_2^2$ is the regularization term that prevents the locally updated model $\boldsymbol{\theta}_{s_c}$ from deviating too far from the server shared parameters $\overline{\boldsymbol{\theta}}_s$.

Let $\ell_c = \mathcal{L}_c(f(\mathbf{x}^{K_c}; \boldsymbol{\theta}_c, \boldsymbol{\theta}_s), y)$ , then the global loss is taken as an average of the loss of each client: $\ell = \sum_{c \in \mathcal{C}} p_c \ell_c$, where $p_c \geq 0$ and $\sum_c p_c = 1$.

The local update is performed using vanilla gradient descent with a local learning rate $\eta$ in each client, and $\boldsymbol{\Theta}_c(r) \in \mathcal{E}$ represents the weight parameters of the client $c$ in the round $r$. Then, for global round $r$,

$$\Delta\boldsymbol{\Theta}_c^{(r)} = \boldsymbol{\Theta}_c^{(r+1)} - \boldsymbol{\Theta}_c^{(r)} = -\eta\left(\nabla\ell_c(\boldsymbol{\Theta}^{(r)}) + 2\lambda\left(\boldsymbol{\theta}_{s_c} - \hat{\boldsymbol{\theta}}_s\right)\right).$$

To better calculate the difference between personalized parameters and shared parameters, we let

$$\boldsymbol{\Theta}_c^{(r)} = \boldsymbol{\theta}_c^{(r)} + \boldsymbol{\theta}_s^{(r)}$$

, where, $\boldsymbol{\theta}_c^{(r)} = [m^{(r)} \quad \mathbf{o}]$, $\boldsymbol{\theta}_s^{(r)} = [\mathbf{o} \quad \mathbf{M}^{(r)}]$.

Specifically, the global aggregation procedure is conducted by taking the average of local updates of shared parameters $\boldsymbol{\theta}_s$ of all $|\mathcal{C}|$ clients. According to

$$\boldsymbol{\theta}_s^{(r+1)} = \overline{\boldsymbol{\theta}}_s^{(r)} = \sum_{c \in \mathcal{C}} \frac{|\mathcal{D}_c|}{N}\boldsymbol{\theta}_{s_c}^{(r)} = \sum_{c \in \mathcal{C}} p_c \boldsymbol{\theta}_{s_c}^{(r)}$$

We make the following standard Assumption commonly used in non-convex optimization (Li et al., 2020b; Reddi et al., 2020).

**Assumption 2** (L-smoothness). $\forall_{c \in \mathcal{C}} \ell_c$ *are L-smooth: for all* $\boldsymbol{\Theta}_1 \in \mathbb{E}$ *and* $\boldsymbol{\Theta}_2 \in \mathbb{E}$,

$$\ell_c(\boldsymbol{\Theta}_1) \leq \ell_c(\boldsymbol{\Theta}_2) + (\boldsymbol{\Theta}_1 - \boldsymbol{\Theta}_2)^T\nabla\ell_c(\boldsymbol{\Theta}_2) + \frac{L}{2}\|\boldsymbol{\Theta}_1 - \boldsymbol{\Theta}_2\|_2^2.$$

**Assumption 3** (Bounded Gradients). *The function* $\ell_c(\boldsymbol{\Theta})$ *have G-bounded gradients, i.e., for any* $c \in \mathcal{C}$, $\boldsymbol{\Theta} \in \mathbb{R}^d$ *we have* $\|\nabla\ell_c(\boldsymbol{\Theta})\| \leq G$.

**Lemma 9** (Smooth Decent Lemma). *Let* $\ell : \mathcal{E} \to \mathbb{R}$ *be an L-smooth function. Then for any* $\boldsymbol{\Theta}^{(r)}, \boldsymbol{\Theta}^{(r+1)} \in \mathbb{E}$, *the following inequality holds:*

$$\ell(\boldsymbol{\Theta}^{(r+1)}) \leq \ell(\boldsymbol{\Theta}^{(r)}) + \langle\nabla\ell(\boldsymbol{\Theta}^{(r)}), \Delta\boldsymbol{\Theta}^{(r)}\rangle + \frac{L}{2}\|\Delta\boldsymbol{\Theta}^{(r)}\|^2.$$

Let $\delta^{(r)} = 2\lambda \sum_{c \in \mathcal{C}} \frac{|\mathcal{D}_c|}{N}\left(\boldsymbol{\theta}_{s_c} - \overline{\boldsymbol{\theta}}_s\right)$. Based on Lemma 1, we have

$$\ell(\boldsymbol{\Theta}^{(r+1)}) \leq \ell(\boldsymbol{\Theta}^{(r)}) + \langle \nabla\ell(\boldsymbol{\Theta}^{(r)}), \Delta\boldsymbol{\Theta}^{(r)} \rangle + \frac{L}{2}\|\Delta\boldsymbol{\Theta}^{(r)}\|^2$$

$$= \ell(\boldsymbol{\Theta}^{(r)}) + \left\langle \nabla\ell(\boldsymbol{\Theta}^{(r)}), -\eta\left(\nabla\ell(\boldsymbol{\Theta}^{(r)}) + \delta^{(r)}\right) \right\rangle$$

$$+ \frac{L\eta^2}{2}\|\nabla\ell(\boldsymbol{\Theta}^{(r)}) + \delta^{(r)}\|^2$$

$$= \ell(\boldsymbol{\Theta}^{(r)}) - \eta\left\langle \nabla\ell(\boldsymbol{\Theta}^{(r)}), \nabla\ell(\boldsymbol{\Theta}^{(r)}) + \delta^{(r)} \right\rangle$$

$$+ \frac{L\eta^2}{2}\|\nabla\ell(\boldsymbol{\Theta}^{(r)}) + \delta^{(r)}\|^2$$

$$= \ell(\boldsymbol{\Theta}^{(r)}) - \eta\|\nabla\ell(\boldsymbol{\Theta}^{(r)})\|^2 - \eta\left\langle \nabla\ell(\boldsymbol{\Theta}^{(r)}), \delta^{(r)} \right\rangle$$

$$+ \frac{L\eta^2}{2}\|\nabla\ell(\boldsymbol{\Theta}^{(r)})\|^2 + L\eta^2\langle\nabla\ell(\boldsymbol{\Theta}^{(r)}, \delta^{(r)})\rangle + \frac{L\eta^2}{2}\|\delta^{(r)}\|^2 \qquad (29)$$

$$= \ell(\boldsymbol{\Theta}^{(r)}) + (\frac{L\eta^2}{2} - \eta)\|\nabla\ell(\boldsymbol{\Theta}^{(r)})\|^2 + \frac{L\eta^2}{2}\|\delta^{(r)}\|^2$$

$$+ (L\eta^2 - \eta)\left\langle \nabla\ell(\boldsymbol{\Theta}^{(r)}), \delta^{(r)} \right\rangle$$

$$= \ell(\boldsymbol{\Theta}^{(r)}) + (\frac{L\eta^2}{2} - \eta)\|\nabla\ell(\boldsymbol{\Theta}^{(r)})\|^2 + \frac{L\eta^2}{2}\|\delta^{(r)}\|^2$$

$$+ \frac{L\eta^2 - \eta}{2}\left(\|\nabla\ell(\boldsymbol{\Theta}^{(r)})\|^2 + \|\delta^{(r)}\|^2 - \|\nabla\ell(\boldsymbol{\Theta}^{(r)}) + \delta^{(r)}\|^2\right)$$

$$= \ell(\boldsymbol{\Theta}^{(r)}) + (L\eta^2 - \frac{3\eta}{2})\|\nabla\ell(\boldsymbol{\Theta}^{(r)})\|^2 + (L\eta^2 - \frac{\eta}{2})\|\delta^{(r)}\|^2$$

$$- \frac{L\eta^2 - \eta}{2}\|\nabla\ell(\boldsymbol{\Theta}^{(r)}) + \delta^{(r)}\|^2$$

We select $\eta = \frac{1}{L}$, so we we have

$$\ell(\boldsymbol{\Theta}^{(r+1)}) \leq \ell(\boldsymbol{\Theta}^{(r)}) - \frac{1}{2L}\|\nabla\ell(\boldsymbol{\Theta}^{(r)})\|^2 + \frac{1}{2L}\|\delta^{(r)}\|^2 \qquad (30)$$

Rearrange the above inequality and we have

$$\|\nabla\ell(\boldsymbol{\Theta}^{(r)})\|^2 \leq 2L\left(\ell(\boldsymbol{\Theta}^{(r+1)}) - \ell(\boldsymbol{\Theta}^{(r)})\right) + \|\delta^{(r)}\|^2 \qquad (31)$$

Then, sum $r$ from 1 to $T$, we have

$$\min_{r\in[T]}\|\nabla\ell(\boldsymbol{\Theta}^{(r)})\| \leq \frac{2L\left(\ell(\boldsymbol{\Theta}^{(r+1)}) - \ell(\boldsymbol{\Theta}^{(r)})\right)}{T} + \frac{1}{T}\sum_{r\in[T]}\|\delta^{(r)}\|^2 \qquad (32)$$

**Definition 7** ($B$-local dissimilarity). *The local functions $\ell_c$ are $B$-locally dissimilar at $\boldsymbol{\Theta}$ if*
$$\mathbb{E}_c[\|\nabla\ell_c(\boldsymbol{\Theta})\|^2] \leq \|\nabla\ell(\boldsymbol{\Theta})\|^2 B^2.$$

*We further define $B(\boldsymbol{\Theta}) = \sqrt{\frac{\mathbb{E}_c[\|\nabla\ell_c(\boldsymbol{\Theta})\|^2]}{\|\nabla\ell(\boldsymbol{\Theta})\|^2}}$ for $\|\nabla\ell(\boldsymbol{\Theta})\| \neq 0$.*

**Definition 8** ($\gamma$-inexact solution). *For a function $h(w; w_0) = F(w) + \lambda\|w - w_0\|^2$, and $\gamma \in [0, 1]$, we say $w^*$ is a $\gamma$-inexact solution of $\min_w h(w; w_0)$ if $\|\nabla h(w^*; w_0)\| \leq \gamma\|\nabla h(w_0; w_0)\|$, where $\nabla h(w; w_0) = \nabla F(w) + \mu(w - w_0)$, where, $\mu = 2\lambda$. Note that smaller $\gamma$ corresponds to higher accuracy.*

Using the notion of $\gamma$-inexactness for each local client, we can define $e_c^{(r)}$ such that
$$\nabla\ell_c\left(\boldsymbol{\Theta}_c^{(r+1)}\right) + \mu\left(\hat{\boldsymbol{\theta}}_s^{(r)} - \boldsymbol{\theta}_{s_c}^{(r)}\right) + \mu\left(\boldsymbol{\theta}_c^{(r+1)} - \boldsymbol{\theta}_c^{(r)}\right) - e_c^{(r)} = 0,$$
$$\|e_c^{(r)}\| \leq \gamma\|\nabla\ell_c\left(\boldsymbol{\Theta}_c^{(r)}\right)\|. \qquad (33)$$

Then we have

$$\boldsymbol{\theta}_s^{(r+1)} - \boldsymbol{\theta}_s^{(r)} = \frac{-1}{\mu}\mathbb{E}_c\left[\nabla\ell_c\left(\boldsymbol{\Theta}_c^{(r)}\right)\right] + \frac{1}{\mu}\mathbb{E}_c[e_c^{(r)}] - \mathbb{E}_c\left[\Delta\boldsymbol{\theta}_c^{(r)}\right], \tag{34}$$

According to (Li et al., 2020a) and triangle inequality, when a regularization is incorporated, ($\lambda > 0$), we have

$$\frac{1}{4\lambda^2}\|\delta^{(r)}\|^2 \leq \left(\mathbb{E}_c\left[\|\boldsymbol{\theta}_s^{(r+1)} - \boldsymbol{\theta}_{s_c}^{(r)}\|\right]\right)^2$$

$$\leq \left(\frac{1+\gamma}{\bar{\mu}}\right)^2\left(\mathbb{E}_c\left[\|\nabla\ell_c\left(\boldsymbol{\Theta}_c^{(r)}\right) - \Delta\boldsymbol{\theta}_c^{(r)}\|\right]\right)^2$$

$$\leq \left(\frac{1+\gamma}{\bar{\mu}}\right)^2\left(\mathbb{E}_c\left[\|\nabla\ell_c\left(\boldsymbol{\Theta}_c^{(r)}\right) - \Delta\boldsymbol{\theta}_c^{(r)}\|^2\right]\right)$$

$$\leq \frac{B^2(1+\gamma)^2}{\bar{\mu}^2}\mathbb{E}\left[\|\nabla\ell_c\left(\boldsymbol{\Theta}_c^{(r)}\right)\|^2\right] + C,$$

Based on the assumption of the bounded gradients (Assumption 3), we find that the $\delta^{(r)}$ is also bounded. Specifically, $C = \left(\frac{1+\gamma}{\bar{\mu}}\right)^2\mathbb{E}_c[\|\Delta\boldsymbol{\theta}_c\|^2] \approx \left(\frac{1+\gamma}{\bar{\mu}}\right)^2\mathbb{E}[\|\Delta M_c\|^2]$. $\|\delta^{(r)}\|^2$ measures the degree of data heterogeneity.

Overall, when $\lambda = 0$, the term $\delta^{(r)} = 0$, eliminating the impact of data heterogeneity and resulting in a convergence rate of $O\left(\frac{1}{T}\right)$, consistent with FedAvg. And when incorporating regularization ($\lambda > 0$), we establish that $\|\delta^{(r)}\|^2$ is bounded, analogous to the theoretical guarantees provided by FedProx (Li et al., 2020a).

This analysis suggests that FlatLand can balance the trade-off between computational overhead and model effectiveness, showing the scalability for the increase in clients. While it introduces additional operations in local computations, these overheads are limited and offer significant optimization opportunities through pre-computation and caching strategies. The method compensates for these minimal costs through reduced communication overhead and enhanced representation capabilities in the Lorentz space, making it a practical and efficient choice for personalized federated learning applications.

## D.5 PERSPECTIVES ON LORENTZ TRANSFORMATIONS

Lorentz Boosts and Lorentz Rotations (Appendix B.3) are understood as transformations that are encapsulated by $\text{LT}\left(\mathbf{x}; \hat{\mathbf{M}}\right)$ when the dimension is unchanged (Chen et al., 2021). We can easily prove that the Lorentz transformations are still covered by $\text{LT}\left(\cdot; \Phi\left(\hat{\mathbf{M}}, \mathbf{N}\right)\right)$, where $\hat{\mathbf{M}} \in \mathbb{R}^{(n+1)\times(n+1)}$, $\mathbf{N} \in \mathbb{R}^{n\times n}$. For any data point $\mathbf{x} \in \mathcal{D}_c$, transformations $\text{LT}\left(\mathbf{x}; \hat{\mathbf{M}}\right)$ and $\text{LT}\left(\mathbf{x}; \Phi\left(\hat{\mathbf{M}}, \mathbf{N}\right)\right)$ map $\mathbf{x}$ to a new spacetime position, maintaining the spacetime interval invariant (Corollary 1), thus preserving the physical and geometric relationships within the same client, in line with special relativity. However, clients with varying spacetime curvatures maintain **distinct spacetime intervals**, reflecting differing underlying data distributions. Moreover, according to the definition of Lorentz Rotation in Equation (8), the server updates only the $\mathbf{M}$, leaving the time-like dimension unchanged. This operation is a relaxation of the Lorentz rotation, consistent with our "Flatland" assumption that aggregates only spatial dimension information.

Table 5: Statistics of graph classification datasets. We report the (average) number of graphs, nodes, edges, classes, and node features of each dataset.

| Dataset | CHEM | | | | | | | BIO | | | SN | | |
|---|---|---|---|---|---|---|---|---|---|---|---|---|---|
| | MUTAG | BZR | COX2 | DHFR | PTC_MR | AIDS | NCI1 | ENZYMES | DD | PROTEINS | COLLAB | IMDB-BINARY | IMDB-MULTI |
| # Graphs | 188 | 405 | 467 | 467 | 344 | 2000 | 4110 | 600 | 1178 | 1113 | 5000 | 1000 | 1500 |
| Avg. # Nodes | 17.93 | 35.75 | 41.22 | 42.43 | 14.29 | 15.69 | 29.87 | 32.63 | 284.32 | 39.06 | 74.49 | 19.77 | 13.00 |
| Avg. # Edges | 19.79 | 38.36 | 43.45 | 44.54 | 14.69 | 16.20 | 32.30 | 62.14 | 715.66 | 72.82 | 2457.78 | 96.53 | 65.94 |
| # Classes | 2 | 2 | 2 | 2 | 2 | 2 | 2 | 6 | 2 | 2 | 3 | 2 | 3 |
| Node Features | original | original | original | original | original | original | original | original | original | original | degree | degree | degree |

Table 6: Statistics of homophilic node classification datasets. We report the (average) number of nodes, edges, classes, clustering coefficient, and heterogeneity for different numbers of clients.

| Dataset | Cora | | | Citeseer | | | ogbn-arxiv | | | Amazon-Photo | | |
|---|---|---|---|---|---|---|---|---|---|---|---|---|
| # Clients | 1 | 10 | 20 | 1 | 10 | 20 | 1 | 10 | 20 | 1 | 10 | 20 |
| # Classes | | 7 | | | 6 | | | 40 | | | 8 | |
| Avg. # Nodes | 2,485 | 249 | 124 | 2,120 | 212 | 106 | 169,343 | 16,934 | 8,467 | 7,487 | 749 | 374 |
| Avg. # Edges | 10,138 | 891 | 422 | 7,358 | 675 | 326 | 2,315,598 | 182,226 | 86,755 | 238,086 | 19,322 | 8,547 |
| Avg. Clustering Coefficient | 0.238 | 0.259 | 0.263 | 0.170 | 0.178 | 0.180 | 0.226 | 0.259 | 0.269 | 0.410 | 0.457 | 0.477 |
| Heterogeneity | N/A | 0.606 | 0.665 | N/A | 0.541 | 0.568 | N/A | 0.615 | 0.637 | N/A | 0.681 | 0.751 |

# E EXPERIMENTAL SUPPLEMENTARY

## E.1 DATASETS

For federated node classification, we adopt four benchmark datasets constructed by (Baek et al., 2023): Cora, CiteSeer, ogbn-arxiv, and Photo (Sen et al., 2008; Hu et al., 2020; Shchur et al., 2018). Cora, CiteSeer, and ogbn-arxiv are citation graphs. Photo is a product graph. Each graph dataset is divided into a certain number of disjoint subgraphs using the METIS graph partitioning algorithm (Karypis & Kumar, 1995), where each subgraph belongs to an FL client. Statistics of datasets are summarized in Table 6.

For federated graph classification, we consider the non-IID settings proposed by (Xie et al., 2021). In total, there are 13 graph classification datasets from three different domains, including small molecules (MUTAG, BZR, COX2, DHFR, PTC_MR, AIDS, NCI1) denoted as CHEM, bioinformatics (ENZYMES, DD, PROTEINS) denoted as BIO, and social networks (COLLAB, IMDB-BINARY, IMDB-MULTI) (Morris et al., 2020) denoted as SN. To simulate data heterogeneity, three non-IID settings are constructed: (1) a cross-dataset setting based on the small molecule datasets (CHEM), (2) a cross-domain setting based on all datasets (BIO-CHEM-SN). In each setting, one dataset corresponds to one FL client. Statistics of datasets are summarized in Table 5 and Table 7.

## E.2 IMPLEMENTATION DETAILS

**Implementation of learnable curvature.** $K$ is a learnable scalar parameter. To ensure the curvature remains negative (as required for hyperbolic space), we implement it as $\text{sigmoid}(K) + 0.5$. This design also keeps curvature $-K$ within an effective range of $[0.5, 1.5]$, which prior work has shown to be ideal for hyperbolic models (Chen et al., 2021). Additionally, this approach maintains numerical stability while satisfying the need for a heterogeneous space.

**Implementation of node classification / graph classification task.** For the node classification task, we employ 2-layer GCN (Kipf & Welling, 2017) for Euclidean models, 2-layer LGCN (Chen et al., 2021) for FlatLand, and HGCN with node selection for FedHGCN (Du et al., 2024). LGCN serves as the backbone for our graph learning framework, combining Lorentz linear layers with graph aggregation operations, similar to how Euclidean counterparts like GCN and GIN integrate linear layers with graph aggregation. Each layer applies a Lorentz transformation followed by neighbor aggregation using the adjacency matrix to get the node representations. We conduct 100 rounds for Cora/CiteSeer and 200 rounds for larger datasets like Photo/ogbn-arxiv, with 1-3 local epochs, use 128-dim hidden layers. For graph classification, we use 3-layer GIN (Xu et al., 2018) as the Euclidean encoder, and the same 3-layer hyperbolic encoders as node classification for hyperbolic models, with 1 local epoch and 200 rounds. The learning rate is chosen from $\{0.01, 0.001\}$, and weight decay uses $1e-5$. We optimize with Adam, and calculate node-level / graph-level accuracy averaged across

Table 7: Statistics of heterophilic node classification datasets. We report the (average) number of nodes and edges, classes

| Dataset | Roman-empire | | | Minesweeper | | | Tolokers | | | Questions | | |
|---|---|---|---|---|---|---|---|---|---|---|---|---|
| # Clients | 1 | 10 | 20 | 1 | 10 | 20 | 1 | 10 | 20 | 1 | 10 | 20 |
| # Classes | | 18 | | | 2 | | | 2 | | | 2 | |
| Avg. # Nodes | 22,662 | 2326 | 1124 | 10000 | 1008 | 513 | 11,758 | 1138 | 605 | 48,921 | 5037 | 2484 |
| Avg. # Edges | 32,927 | 6682 | 3268 | 39402 | 7696 | 3820 | 519,000 | 40000 | 15000 | 153,540 | 20,000 | 7702 |

| Method | Roman-Empire | Minesweeper | Tolokers | Questions |
|---|---|---|---|---|
| FedAvg (10) | 64.60 | 75.32 | 54.29 | 55.55 |
| **FlatLand (Ours)** | $66.10 \pm 0.21$ | $76.34 \pm 0.05$ | $71.34 \pm 0.06$ | $67.71 \pm 0.08$ |

Table 8

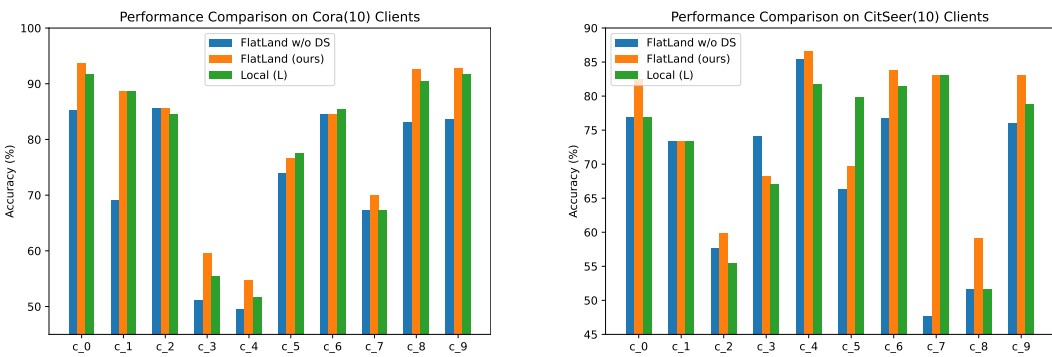

Figure 9: Performance comparison of FlatLand on Cora and CitSeer across local 10 clients.

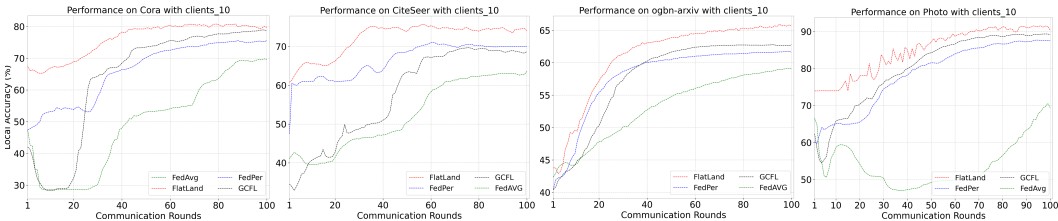

Figure 10: The convergence curves of our proposed methods and the strong baselines.

clients. All experiments are implemented in Python3.10, PyTorch, and run on an RTX A6000 GPU, 40G storage. Each client is allocated a worker with one round of around 1 second for one epoch in the node classification task.

**Backbone Structure.** Inspired by recent GNN models (Chen et al., 2024) that highlight the benefits of incorporating high-pass information for heterophilic graphs, all hyperbolic baselines in our framework combine information from both low-pass (adjacency-based) and high-pass (Laplacian-based) operations through a learnable gating mechanism. For a fair comparison, we also applied this design to the Euclidean baselines, but it did not yield improvements over the hyperbolic counterparts, suggesting that hyperbolic models benefit more from the joint use of low and high pass information than Euclidean ones (Table 8)

These results demonstrate that FlatLand performs competitively on image data. This indicates that FlatLand effectively handles high data heterogeneity and scales well with different numbers of clients. Besides, the significant performance gap between FlatLand and traditional federated learning methods like FedAvg and FedProx highlights the effectiveness of our approach in highly heterogeneous settings.

### E.3    Parial Participation Rate

We conducted extensive experiments with an increased number of clients (50 clients) in the Cora dataset, which represents a large client pool configuration in graph federated learning scenarios (Du et al., 2024). The results demonstrate that our method maintains its effectiveness even with an expanded client base. Furthermore, we investigated the impact of partial client participation, where only a fraction of clients participate in each aggregation round. Figure 11 illustrates the performance comparison among FedAvg and strong baselines (FedHGCN, FedPer) and FlatLand under different participation rates on the Cora dataset with 50 clients. Due to GCFL requiring careful parameter tuning and the difficulty in finding optimal results, combined with its performance on the Cora dataset being generally comparable to FedPer, we have not included it in this experiment.

The experimental results show that FlatLand exhibits remarkable robustness across various participation rates. Even with only 10% client participation (5 clients), FlatLand achieves an accuracy of 81.82%, while FedAvg only reaches 18.14%. As the participation rate increases, FlatLand maintains consistently higher performance than all baselines. In contrast, FedAvg shows performance fluctuations.

These findings confirm that FlatLand can maintain high performance even under low client participation scenarios, demonstrating its practical value for real-world federated learning applications where full client participation may not always be feasible. The robust performance under partial participation is particularly important for federated learning systems, where coordinating all clients simultaneously can be challenging.

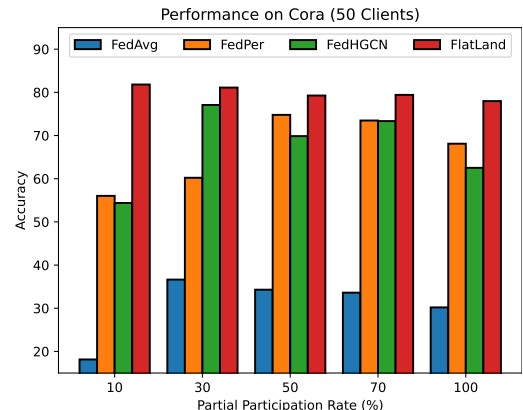

Figure 11: Performance comparison between FedAvg and FlatLand under different client participation rates on Cora dataset with 50 clients.

### E.4    Impact of Curvature Initialization

We further investigate the impact of different curvature initialization strategies on model performance. Specifically, we compare four initialization methods: (1) **Forman-Ricci curvature**, (2) **Ollivier-Ricci curvature**, (3) **constant-**1, and (4) an **MLP-based** estimator that updating the curvature via MLP layer. Table 9 reports the results on the CiteSeer dataset with 10 and 20 clients.

Table 9: Performance with different curvature initialization methods on CiteSeer. Results are reported as mean $\pm$ standard deviation over five runs.

| Clients | Ricci | Ollivier | Const. | MLP |
|---|---|---|---|---|
| 10 | $73.90 \pm 0.23$ | $73.51 \pm 0.18$ | $72.91 \pm 0.20$ | $72.96 \pm 0.46$ |
| 20 | $72.24 \pm 0.24$ | $72.21 \pm 0.26$ | $71.89 \pm 0.31$ | $72.59 \pm 0.50$ |

As shown in Table 9, the choice of initialization method has only a marginal effect on performance. This indicates that our method rubust to the initialization is *not critical* to the effectiveness of our method. What truly matters is that each client is assigned a **learnable curvature**, which can be adapted during training to better fit its local data distribution (see Figure 5).

### E.5    Convergence Curves

The convergence curves are shown in Figure 10. As the figures demonstrate, our proposed method can achieve better convergence speed, highlighting the superiority of our proposed approach.

## F    USE OF LLMS

This work utilizes large language models (LLMs), including ChatGPT https://chat.openai.com/ and Claude Code https://claude.ai/chats, as general-purpose tools for writing assistance. These tools were employed for polishing text, correcting formatting errors, and checking grammar throughout the writing process.

This use of LLMs complies with the ICLR 2026 Author Guide https://iclr.cc/Conferences/2026/AuthorGuide, ensuring adherence to the conference's guidelines for AI assistance in academic writing.

