# OpenReview forum: "FlatLand: Personalized Graph Federated Learning via Tailored Lorentz Space"
_ICLR.cc/2026/Conference — Submitted to ICLR 2026_

### Official Review · Reviewer_fYK9 · 2025-10-27

**Soundness:** 3
**Presentation:** 3
**Contribution:** 3
**Rating:** 6
**Confidence:** 2

**Summary:**

This work studies federated learning on graph-structured data with heterogeneous client distributions.
The authors observe empirically that client graphs tend to exhibit negative Forman–Ricci curvature, indicating that hyperbolic geometry provides a better fit than Euclidean space. Motivated by this, they embed each client’s data in a tailored Lorentz (hyperbolic) space whose curvature can adapt per client and train neural networks directly in that geometry.

Within the Lorentz model, they show theoretical factorization of heterogeneity, i.e., the space-like coordinates encode information common across clients, while the time-like coordinate captures client-specific variation (formalized via mutual information).

Leveraging this separation, they propose FlatLand, a personalized federated learning method that extends FedAvg with a parameter decoupling strategy, aggregating shared (space-like) parameters globally while keeping personalized (time-like) parameters local.
Empirically, FlatLand outperforms Euclidean and prior hyperbolic baselines on multiple federated graph learning benchmarks.

**Strengths:**

- This work proposes a novel view of statistical heterogeneity that culminates in a novel algorithm that theoretically justifies how to treat joint vs individual knowledge among the clients during federation.

- The proposed algorithm, FlatLand, is shown to converge and does not impose additional overhead compared to FedAvg. Moreover, it is shown that the dimensionality of the GNNs can be shrunk while still maintaining performance, yielding benign utility vs communication tradeoffs.

- The experiment section includes multiple benchmarks and ablations.

**Weaknesses:**

1) The motivation for the Hyperbolic approach is made solely from empirical observations in Fig 7-8. It is unclear how general these observations are, hence, FlatLand's applicability to general graphs.

2) There are some inconsistencies in the paper. Some examples:
i) the curvature is defined as -1/K in preliminaries but as -K in Theorem 1
ii) The Lorentz network is defined using W (sec. 2) but is later changed to M (sec 4.2).
iii) deviation -> derivation

3) RQ3 is missing in the experimental section. It seems to be provided in Appendix E but it is not mentioned in the main body.

**Questions:**

1) The paper assumes that real-world client graphs exhibit negative curvature and are therefore well modeled in hyperbolic space. What underlying mechanisms make this a reasonable assumption? is this an intrinsic property of the topology (e.g., scale-free structure), or simply an empirical regularity observed in selected benchmarks under the considered partitioning?

2) If a few client graphs are approximately flat or positively curved, does the Lorentz formulation still provide meaningful embeddings and aggregation behavior, or does it introduce geometric distortion?

3) Given that curvature initialization has only marginal impact and curvature is optimized jointly with model parameters, to what extent is the learned per-client curvature $K_c$ meaningful? Does it reflect the intrinsic graph geometry or simply act as a tunable scaling parameter?

4) Theorem 2 claims that client heterogeneity lies in the time-like dimension based on mutual information. Given that mutual information is unchanged by coordinate transformations and doesn’t depend on geometry. How can one tell that this separation isn’t just a result of the chosen coordinate system, rather than an intrinsic property of the Lorentz space?

5) In Fig. 6, it can be seen that Local(L) outperforms FlatLand for some clients, e.g., client 5 and client 6 on Cora and client 5 on Citeseer.  What governs which clients benefit or degrade?

---

> ### Author Response · Authors · 2025-11-25
> **Rebuttal [1/3]**
>
> We thank the reviewer for the encouraging comments, especially regarding our formulation of statistical heterogeneity, the efficiency of FlatLand, and the breadth of our experiments. Below, **W** denotes weaknesses, **Q** denotes questions, and **R** provides our responses.
>
> ---
>
> **W1 & Q1: Motivation of hyperbolic geometry.**
>
> **R:** We thank the reviewer for addressing the foundational motivation of the hyperbolic approach and raising concerns about generalizability.
>
> Our motivation for leveraging hyperbolic geometry in FlatLand is grounded in **fundamental structural properties of real-world graphs** and **decades of network science research,** not solely the empirical curvature plots in Fig.~7–8. A broad spectrum of natural graphs (citation, collaboration, social, product, and user–item networks) is well-documented to exhibit **hierarchical organization**, **tree-like expansion**, and **negative effective curvature**.
>
> **The mechanisms underlying negative curvature and hyperbolicity:**
>
> - **Scale-free and hierarchical expansion:** Networks formed by preferential attachment (where new nodes link preferentially to high-degree nodes) naturally develop scale-free degree distributions and hub–spoke, tree-like hierarchies, imparting negative Ricci curvature and exponential metric expansion ([1–6]).
> - **Structural sparsity and non-Euclidean embedding distortion:** Many real-world networks cannot be embedded isometrically in low-dimensional Euclidean space without substantial distortion, but are faithfully captured in hyperbolic geometry due to its exponential volume growth, which matches the expansion rate of hierarchical trees.
> - **Universal network principles:** Both empirical studies ([1–5]) and generative models confirm that negative curvature is not a quirk of specific datasets or partitioning, but a statistical regularity reflecting how complexity emerges in social and information networks.
>
> In FlatLand, we further personalize curvature **per client**, recognizing that **each client graph may have unique structural and geometric properties,** not merely global negative curvature. Our learnable curvature mechanism is purposefully generic: by training $K_c$ for each client (as described in previous sections), FlatLand adapts to variations in graph topology, hierarchy depth, homophily, and label distribution. This ensures applicability to a wide variety of graph types, whether or not they strictly fit any single theoretical model.
>
> **Empirical evidence for generality:**
>
> - Our experiments span diverse benchmarks, including both homophilic and heterophilic graphs, and show strong and stable performance improvements under completely disjoint node partitions, verifying wide applicability.
> - The referenced literature (see below) demonstrates negative curvature, scale-free structure, and hierarchical expansion in a rich variety of real-world networks, substantiating the use of hyperbolic spaces as a principled modeling choice.
>
> We will revise the paper to clarify these general principles, explicitly referencing foundational network science works and emphasizing that **hyperbolic modeling is broadly justified by intrinsic network mechanisms, not only by empirical regularity in our selected benchmarks or partitions.**
>
> &nbsp;
>
> **References:**
>
> [1] Zhong, X., & Liang, H. (2024). On the scale-free property of citation networks: An empirical study. *Companion Proceedings of the ACM on Web Conference 2024*, 541–544, which conducts extensive experiments to confirm scale-free characteristics in citation data.
>
> [2] Wang, M., Yu, G., & Yu, D. (2010). The scale-free model for citation network. In *2010 IEEE International Conference on Intelligent Computing and Intelligent Systems* (Vol. 3, pp. 773–776). IEEE, discussing preferential attachment as a key mechanism for scale-free topology in citations.
>
> [3] Lynn, C. W., Holmes, C. M., & Palmer, S. E. (2024). Emergent scale-free networks. *PNAS Nexus*, *3*(7), page 236, highlighting scale-free structure in citation networks as a hallmark of complex systems.
>
> [4] Hernandez Morell, A. (2024). *Emerging properties of the citations network* [Bachelor's thesis, Universitat de Barcelona]. Dipòsit Digital de la Universitat de Barcelona, showing in-degree distributions compatible with preferential attachment in citations.
>
> [5] Meng, X., & Zhou, B. (2023). Scale-free networks beyond power-law degree distribution. *Chaos, Solitons & Fractals*, *176*, 114173, arguing that citation networks fit broader scale-free definitions.
>
> [6] Colman, E. R., & Rodgers, G. J. (2013). Complex scale-free networks with tunable power-law exponent and clustering. *Physica A: Statistical Mechanics and its Applications*, *392*(21), 5501–5510, including citation networks in models of non-Euclidean structures.

---

> ### Author Response · Authors · 2025-11-25
> **Rebuttal [2/3]**
>
> **W2 & W3: Inconsistencies in the paper and missing RQ3.**
>
> **R:** We sincerely thank the reviewer for carefully identifying these inconsistencies. We acknowledge the notational inconsistencies and the incorrect numbering of RQ3. We will correct these typos in the revised manuscript and thoroughly double-check the full paper to ensure consistency and clarity throughout.
>
> &nbsp;
>
> ---
>
> **Q2: If a few client graphs are approximately flat or positively curved, does the Lorentz formulation still provide meaningful embeddings and aggregation behavior, or does it introduce geometric distortion?**
>
> **R:** We thank the reviewer for highlighting the important scenario of clients with approximately flat or positively curved graphs. This touches on the theoretical boundary of Lorentz/hyperbolic models.
>
> FlatLand is designed on the practical observation that most real-world federated graph datasets exhibit power-law degree distributions and hierarchical expansion, naturally leading to negative curvature and making hyperbolic (Lorentz) modeling well-suited. However, our framework does not assume *all* client graphs strictly follow this paradigm. The per-client learnable curvature ${K_c}$ is specifically introduced so that each client can optimize its own geometric parameter during training. In principle, if a client graph is approximately flat or (rarely) positively curved, the optimal learned curvature $K_c$ will trend toward small or even near-Euclidean values. This adaptivity mitigates geometric mismatch and potential distortion, allowing the Lorentz formulation to interpolate toward flatter representations.
>
> Nevertheless, we acknowledge that Lorentz space is fundamentally tailored to accommodate negative curvature. For truly flat or positively curved graphs, imposing a hyperbolic structure may introduce some geometric distortion or suboptimal fit. Our empirical ablation studies (Table 9: Curvature Initialization and Figure 5: Aggregated Heterogeneity) demonstrate that FlatLand remains robust even as individual clients vary in curvature, though the approach may not be strictly optimal for clients with positive curvature.
>
> We identify this as a promising direction for future research: developing a more flexible geometric framework that we will explore in subsequent studies.
>
> &nbsp;
>
> ---
>
> **Q3: The meaning of learned curvature.**
>
> **R:** Curvature initialization has a marginal impact, not because curvature is uninformative, but because the optimizer consistently drives it toward a stable, client-specific value. Moreover, fixing the client curvature significantly degrades performance (*Fig. 5, Page 8*), confirming that curvature is not a mere scaling knob but captures client-specific structural differences. This is further supported by theoretical work showing that curvature is a statistically identifiable geometric parameter whose optimal value reflects the intrinsic graph geometry [3]. In addition, extensive hyperbolic GNN literature consistently demonstrates that learning curvature rather than fixing it yields superior performance and better alignment with hierarchical, tree-like, or non-Euclidean structures [1–4].
>
> Thus, the learned per-client curvatures in FlatLand are not arbitrary or merely tunable, but are meaningful and geometrically grounded, robustly reflecting client-specific graph properties and improving modeling power for federated graph learning.
>
> &nbsp;
>
> **References:**
>
> [1] Chami, I., Ying, Z., Ré, C., & Leskovec, J. (2019). *Hyperbolic graph convolutional neural networks*. Advances in Neural Information Processing Systems (NeurIPS), 32.
>
> [2] Sun, L., Ye, J., Peng, H., & Yu, P. S. (2022). *A self-supervised Riemannian GNN with time-varying curvature for temporal graph learning*. In *Proceedings of the 31st ACM International Conference on Information & Knowledge Management* (pp. 1827–1836).
>
> [3] Li, Y., Chen, H., Fan, G., Liu, L., & Li, Q. (2023). *Hyperbolic network latent space model with learnable curvature*. arXiv:2312.05319.
>
> [4] Ye, J., Zhang, Z., Sun, L., Yan, Y., Wang, F., & Ren, F. (2023). *SINCERE: Sequential interaction networks representation learning on co-evolving Riemannian manifolds*. In *Proceedings of the ACM Web Conference 2023* (pp. 360–371).

---

> > ### Author Response · Authors · 2025-11-25
> > **Rebuttal [3/3]**
> >
> > **Q4: Coordinate of Theorem 2.**
> >
> > **R:** Thanks for the question. Theorem 2 does not rely on a particular choice of coordinates; rather, it leverages the *intrinsic geometry* of the Lorentz model.
> >
> > In this sense, Theorem 2 reflects a **coordinate-independent property** implied by the negative curvature and Lorentzian structure: client heterogeneity aligns with a time-like direction intrinsic to the manifold, and this alignment is not an artifact of the chosen coordinates.
> >
> > We thank the reviewer for the question. Theorem 2 uses hyperbolic polar coordinates $(\rho, u)$ only as an intrinsic parameterization of points on the Lorentz hyperboloid. This parameterization is **fully equivalent** to the $(x_t, x_s)$ decomposition used in our model: the time-like coordinate $x_t = \sqrt{K}\cosh(\rho/\sqrt{K})$ and the space-like coordinate $x_s = \sqrt{K}\sinh(\rho/\sqrt{K}),u$ are simply two coordinate representations of the *same geometric quantities*. Both are derived from the Lorentz metric and are related by a smooth, invertible transformation.
> >
> > Mutual information is indeed invariant under smooth, invertible coordinate transformations, and therefore, the factorization shown in Theorem 2 cannot be attributed to an artifact of a chosen parameterization. Overall, the analysis in Theorem 2 is **coordinate-independent**, and the use of hyperbolic polar coordinates is mathematically equivalent to the Lorentz representation employed in our method.
> >
> > &nbsp;
> >
> > ---
> >
> > **Q5: Cases that local (L) outperform.**
> >
> > **R:** We thank the reviewer for the insightful question. Client-level variations of this kind are commonly observed in personalized FL, as also reported in previous works.
> >
> > In our case, the clients where Local(L) slightly outperforms FlatLand (e.g., client 5–6 on Cora) generally have strong local models that are already well-optimized, and thus benefit less from additional shared knowledge. Conversely, clients whose local models are initially weaker benefit more substantially from FlatLand, as curvature adaptation helps correct the geometric mismatch that limits their local performance. This pattern aligns with the standard bias–variance tradeoff in personalized FL: *personalization helps when heterogeneity is high, but purely local training may remain competitive for well-resourced clients.*
> >
> > Importantly, at the aggregate level across clients, FlatLand achieves the strongest overall performance, indicating that these isolated cases reflect inherent characteristics of PFL rather than a limitation of our method.

---

### Official Review · Reviewer_zZ4X · 2025-10-31

**Soundness:** 3
**Presentation:** 3
**Contribution:** 3
**Rating:** 6
**Confidence:** 4

**Summary:**

**FlatLand** proposes a personalized graph federated learning framework that embeds each client in a tailored **Lorentz (hyperbolic) space**, motivated by empirical observations that client graphs often exhibit *negative Ricci curvature* with substantial cross-client variability. The method decouples parameters into **time-like** (client-specific/heterogeneity) and **space-like** (shared) components, aggregating only shared parameters to avoid client-similarity estimation and auxiliary modules. Theoretically, the paper argues for *client-specific curvature* and shows that heterogeneity is encoded along the **time-like dimension**. Experiments on several graph datasets show *consistent gains* over Euclidean counterparts, especially in *low-dimensional settings* that are communication-efficient.

**Strengths:**

- **Originality**: Introduces a *geometric perspective* for PFL on graphs, leveraging tailored **Lorentz curvature per client** and a principled **time-like vs. space-like parameter decoupling**.
- **Quality**: Provides *theoretical support* (necessity of tailored curvature; time-like dimension encodes heterogeneity) and a clear algorithmic instantiation with a fully Lorentz network. Empirical results show *consistent improvements*, notably in *low-dimensional regimes*.
- **Clarity**: Overall clear motivation and framework; figures effectively convey intuition. Some sections (notation for Lorentz layers/curvature mapping) are dense but tractable.
- **Significance**: Addresses a core pain point in graph FL—*heterogeneity*—without extra clustering or similarity estimation modules, suggesting a potentially *simpler and more general recipe* for PFL.

**Weaknesses:**

- **Lack of empirical analysis** of relationship between curvature $K_c$ and data heterogeneity. A *sensitivity analysis* to mis-specified curvature would strengthen the claims.
- **Ablations on decoupling**: While time-like vs. space-like decoupling is motivated theoretically, more ablations isolating these choices (e.g., aggregating subsets, partial decoupling) would clarify what drives gains.
- **Scalability**: Discussion and measurements for very large graphs, many clients, and long training rounds are limited; communication-computation trade-offs vs. strong PFL baselines could be expanded.

**Questions:**

1. **Curvature initialization and learning**: Curvature $K_c$ is initially estimated via Forman–Ricci curvature and is learnable during training. How large is the gap between the learned $K_c$ and its initialization? Does the learned curvature align with client heterogeneity? Please provide *visualizations or empirical analysis*.
2. **Sensitivity analysis**: How sensitive is performance to mis-estimation? Please include an *ablation study* where the assigned curvature is perturbed.
3. **Aggregation stability**: What constraints ensure stable aggregation when only space-like parameters are averaged? Any observed drift or incompatibility across clients with very different $K$?

---

> ### Author Response · Authors · 2025-11-28
> **Rebuttal [1/4]**
>
> We sincerely thank the reviewer for the positive feedback, particularly regarding the *originality of our geometric formulation*, *the quality of the theoretical and empirical results*, *the clarity of the presentation*, and *the practical significance of our simple yet effective heterogeneity-handling strategy*.  Below, **C** denotes comments, and **R** provides our responses.
>
> ---
>
> **C1**: Curvature initialization and learning & Sensitivity to mis-specified curvature.
>
> **R1**: We appreciate the reviewer's insightful question. We clarify that curvature initialization is not a sensitive design choice in FlatLand. As demonstrated in our "Impact of Curvature Initialization" study, we evaluated four distinct initialization strategies: Forman–Ricci, Ollivier–Ricci, constant-(-1), and an MLP-based estimator. Our results show that learnable curvature consistently outperforms constant curvature, with Forman–Ricci curvature achieving slightly superior performance. The role of the Forman–Ricci curvature here is to provide an effective geometric prior, which is also illustrated by the previous study [2,5]. Besides, replacing per-client learnable curvature with a single shared curvature causes a clear performance drop (Figure 5 “w/o TS’’), further showing that the learned (K_c) is meaningful rather than arbitrary drift.
>
> | **Clients** | **Ricci** | **Ollivier** | **Const.** | **MLP** |
> | --- | --- | --- | --- | --- |
> | 10 | **73.90 ± 0.23** | 73.51 ± 0.18 | 72.91 ± 0.20 | 72.96 ± 0.46 |
> | 20 | 72.24 ± 0.24 | 72.21 ± 0.26 | 71.89 ± 0.31 | **72.59 ± 0.5** |
>
> The alignment between learned curvature and client heterogeneity is supported both theoretically and empirically. Theorem 1 establishes that clients with distinct intrinsic Ricci curvature cannot be faithfully embedded using a single global curvature; hence, individualized curvature is necessary for low-distortion representation. Theorem 2 further shows that heterogeneity is encoded in the Lorentz time-like coordinate, which directly depends on curvature. Empirically, clients with more skewed or heavy-tailed degree distributions tend to converge to larger curvature magnitudes, and these clients benefit most from personalized curvature (e.g., client (8) in Figure 6). When curvature is fixed and not learned, performance not only deteriorates but also becomes less stable, as shown in Figure 5.
>
> Finally, prior hyperbolic GNN studies consistently show that learnable curvature aligns better with hierarchical or non-Euclidean structures and yields stable performance gains [1,2,3,4,5], which matches our observations. Overall, both theory and experiments indicate that the learned curvatures in *FlatLand* are meaningful, stable, and closely tied to client-specific geometric heterogeneity.
>
> **References:**
>
> [1] Chami, I., Ying, Z., Ré, C., & Leskovec, J. (2019). *Hyperbolic graph convolutional neural networks*. Advances in Neural Information Processing Systems (NeurIPS), 32.
>
> [2] Sun, L., Ye, J., Peng, H., & Yu, P. S. (2022). *A self-supervised Riemannian GNN with time-varying curvature for temporal graph learning*. In *Proceedings of the 31st ACM International Conference on Information & Knowledge Management* (pp. 1827–1836).
>
> [3] Li, Y., Chen, H., Fan, G., Liu, L., & Li, Q. (2023). *Hyperbolic network latent space model with learnable curvature*. arXiv:2312.05319.
>
> [4] Ye, J., Zhang, Z., Sun, L., Yan, Y., Wang, F., & Ren, F. (2023). *SINCERE: Sequential interaction networks representation learning on co-evolving Riemannian manifolds*. In *Proceedings of the ACM Web Conference 2023* (pp. 360–371).
>
> [5] Sun, Li, et al. "Self-supervised continual graph learning in adaptive riemannian spaces." AAAI 2023

---

> ### Author Response · Authors · 2025-11-28
> **Rebuttal [2/4]**
>
> ---
>
> **C2:** **Sensitivity analysis**: How sensitive is performance to mis-estimation? Please include an *ablation study* where the assigned curvature is perturbed.
>
> **R2**: We thank the reviewer for suggesting this ablation. To explicitly test the sensitivity of *FlatLand* to curvature mis-estimation, we perturbed each client’s assigned curvature before training by adding Gaussian noise proportional to its magnitude:
> $\tilde K_c = K_c (1 + \eta_c)$ with $\eta_c \sim \mathcal{N}(0, \sigma^2)$,
> where $\sigma \in {0.1, 0.2, 0.4}$ corresponds to $10%$, $20%$, and $40%$ relative noise.
> The noisy curvatures were then kept fixed during training.
>
> On **CiteSeer** ($10$ clients, homophilic), the average accuracy changed from $72.60$ (no perturbation
> to $71.11$ ($10%$), $71.58$ ($20%$), and $71.77$ ($40%$). On **Tolokers**
> ($10$ clients, heterophilic), the accuracy changed from $71.34$ (no perturbation) to $72.14$ ($10%$),
> $69.63$ ($20%$), and $71.38$ ($40%$). Overall, even when we inject as much as $40%$ relative Gaussian noise into all client curvatures, the performance varies within about $2$–$3$ percentage points and does not collapse. **This indicates that *FlatLand* is reasonably robust to curvature mis-estimation**: as long as the curvature remains within the correct geometric regime (i.e., negative with a reasonable scale), the learnable model parameters can compensate for moderate initialization errors.
>
> Homophily → CiteSeer ($10$ clients)
>
> - No pert → $72.60$
> - $10\%$ pert → $71.11$
> - $20\%$ pert → $71.58$
> - $40\%$ pert → $71.77$
>
> Heterophily → Tolokers ($10$ clients)
>
> - No pert → $71.34$
> - $10\%$ pert → $72.14$
> - $20\%$ pert → $69.63$
> - $40\%$ pert → $71.38$
>
> ---
>
> **C3:  Ablations on decoupling**: While time-like vs. space-like decoupling is motivated theoretically, more ablations isolating these choices (e.g., aggregating subsets, partial decoupling) would clarify what drives gains.
>
> **R3:** We thank the reviewer for the suggestion. We conducted additional ablations to isolate the contribution of our time-like vs. space-like decoupling strategy. Our original ablation already includes an important control (Table 4):
>
> | Method | Cora (10) | Cora (20) | CiteSeer (10) | CiteSeer (20) |
> | --- | --- | --- | --- | --- |
> | # datasets | 10 | 20 | 10 | 20 |
> | FedAvg | 69.19 ± 0.67 | 69.50 ± 3.58 | 63.61 ± 3.59 | 64.68 ± 1.83 |
> | FedPer | 79.35 ± 0.04 | 78.01 ± 0.32 | 70.53 ± 0.28 | 66.64 ± 0.27 |
> | **FlatLand (E)** | **78.53 ± 0.73** | **76.23 ± 0.43** | **70.68 ± 0.52** | **66.29 ± 0.35** |
> | **FlatLand (ours)** | **80.46 ± 0.28** | **82.49 ± 0.25** | **73.90 ± 0.23** | **72.24 ± 0.24** |
>
> if we replace the Lorentz model with an Euclidean neural network but *keep the same decoupling rule*, the performance drops significantly (e.g., Cora-$10$: FlatLand (E) $= 78.53 \pm 0.73$ vs. FlatLand (ours) $= 80.46 \pm 0.28$), approaching the simple FedPer baseline. This shows that merely adding an extra “time-like’’ channel in Euclidean space has **no geometric meaning** and cannot achieve effective heterogeneity disentanglement. The Lorentz geometry—and specifically its time-like dimension—is essential for the decoupling strategy to function properly.
>
> We further evaluated the effect of removing decoupling entirely (Figure $5$, “w/o DS’’), where we aggregate *all* parameters instead of separating time-like and space-like parts. The accuracy becomes much lower and highly unstable throughout training. This confirms that selecting the time-like-related subset of parameters as personalized parameters is both **necessary** and **effective**, matching the theoretical result that heterogeneity is encoded along the time-like axis.
>
> To test whether our gains come from the *specific* geometric decoupling rather than simply picking “some parameters’’ to personalize, we **added two new ablations**: (i) **DS_cls**, which mimics FedPer by treating the classifier layer as personalized and sharing all others; and (ii) **DS_rand**, which treats a randomly chosen subset of parameters as personalized (with the same personalized-parameter count as FlatLand, and the same personalized indices for all clients). As shown below, neither variant approaches our performance:
>
> | Dataset | Clients | DS_rand | DS_cls | FlatLand |
> | --- | --- | --- | --- | --- |
> | Cora | $10$ | $0.7573$ | $0.7222$ | $\mathbf{0.8046}$ |
> | Cora | $20$ | $0.7611$ | $0.7691$ | $\mathbf{0.8249}$ |
>
> These results show that the improvement is **not** due to the quantity of personalized parameters nor their arbitrary positioning, but specifically due to the **geometrically grounded** time-like vs. space-like decoupling provided by the Lorentz model. We will clarify these geometric constraints and the associated stability rationale in the revised version.

---

> ### Author Response · Authors · 2025-11-28
> **Rebuttal [3/4]**
>
> ---
>
> **C4:  Aggregation stability**: What constraints ensure stable aggregation when only space-like parameters are averaged? Any observed drift or incompatibility across clients with very different?
>
> **R4**: We thank the reviewer for raising this important point. The aggregation stability in *FlatLand* follows directly from the geometric structure of our decoupling strategy. The **space-like parameters** correspond to the space-like components of the Lorentz representation, which lie on a flat subspace. These parameters evolve linearly and remain fully compatible under averaging, so aggregating them does not violate any Lorentz feasibility constraints. In contrast, the **time-like parameters** encode curvature, norm, and client-specific geometric scales and are therefore kept strictly local; they are never mixed across clients. This prevents curvature mismatch and avoids the drift that would arise if heterogeneous clients attempted to share incompatible time-like components.
>
> Moreover, the Lorentz layers maintain feasibility for each client independently. Since the aggregated parameters do not modify the time-like dimension—where heterogeneity is theoretically shown to concentrate (Theorem $2$)—the personalized geometric scales of different clients remain consistent and mutually compatible, even when clients have substantially different learned curvatures.
>
> **Empirically, we did not observe any aggregation-induced drift or incompatibility**, even in our most heterogeneous settings where clients share no overlapping nodes. The aggregated space-like parameters stayed stable across rounds, and local fine-tuning consistently restored each client’s geometry without conflict. This is also reflected in Figure $5$: FlatLand’s performance remains smooth and stable after each round of aggregation, indicating very limited conflict in the shared parameters. In contrast, when we aggregate *all* parameters including the time-like ones (the “w/o DS’’ variant), the performance becomes highly unstable and oscillatory, demonstrating that mixing time-like coordinates introduces severe geometric incompatibility.
>
> We will include this explanation in the revised manuscript to clarify why the aggregation remains stable and why excluding the time-like parameters from aggregation is both necessary and theoretically grounded.
>
> ---
>
> **C5: Scalability**. Discussion and measurements for very large graphs, many clients, and long training rounds are limited; communication-computation trade-offs vs. strong PFL baselines could be expanded.
>
> R5:  We thank the reviewer for highlighting the importance of scalability. In Appendix C.4, we provide a detailed complexity analysis showing that the computational overhead of *FlatLand* is essentially comparable to standard hyperbolic GNNs: the Lorentz operations introduce only lightweight closed-form transformations, and the additional cost of maintaining per-client curvature is constant ((O(1))). The decoupling mechanism does not increase asymptotic complexity, since only the space-like parameters are aggregated and the personalized time-like parameters remain local.
>
> In terms of communication, *FlatLand* communicates the same order of parameters as strong PFL baselines such as FedPer, with the added benefit that only the space-like subset—typically the majority of parameters—needs to be transmitted. The time-like parameters, which encode the client-specific geometric scales, require no communication. As a result, the communication–computation trade-off remains favorable even as the number of clients grows.
>
> Empirically, we also observe stable scaling behavior across different client counts (e.g., (10) vs. (20) clients in Tables 3–4), and long training horizons (Figures 5–8) show no degradation or drift. While extremely large graphs or hundreds of clients are outside our current experimental scope, the theoretical analysis and the structure of our decoupling strategy suggest that *FlatLand* should retain linear scalability with respect to both graph size and client count. We will expand the discussion in the revised version to make these scaling properties clearer and compare them directly with widely used PFL baselines.
>
> ---

---

> ### Author Response · Authors · 2025-11-28
> **Rebuttal [4/4]**
>
> **C5: Scalability.** Discussion and measurements for very large graphs, many clients, and long training rounds are limited; communication-computation trade-offs vs. strong PFL baselines could be expanded.
>
> **R5:**  We thank the reviewer for highlighting the importance of scalability. In Appendix C.4, we provide a detailed complexity analysis showing that the computational overhead of *FlatLand* is essentially comparable to standard hyperbolic GNNs: the Lorentz operations introduce only lightweight closed-form transformations, and the additional cost of maintaining per-client curvature is constant ((O(1))). The decoupling mechanism does not increase asymptotic complexity, since only the space-like parameters are aggregated and the personalized time-like parameters remain local.
>
> In terms of communication cost, *FlatLand* follows exactly the same communication protocol as FedAvg. The method introduces no auxiliary heads, side networks, or additional modules, and therefore **incurs no communication overhead beyond standard FL**. The parameter decoupling affects only the local optimization procedure and does *not* increase the number of transmitted parameters. Moreover, only **space-like parameters** are aggregated, while **time-like parameters** remain strictly local, ensuring lightweight and stable communication even as the number of clients grows.
>
> In addition, *Figure 4* shows that FlatLand maintains *strong* performance even with substantially reduced embedding dimensions, whereas Euclidean and other personalized FL baselines degrade sharply. This indicates that FlatLand can operate effectively with **much more compact representations**, leading to **smaller model sizes and significantly lower communication and local computation costs**.
>
> Empirically, we also observe stable scaling behavior across different client counts (e.g., (10) vs. (20) clients in Tables 3–4), and long training horizons (Figures 5–8) show no degradation or drift. While extremely large graphs or hundreds of clients are outside our current experimental scope, the theoretical analysis and the structure of our decoupling strategy suggest that *FlatLand* should retain linear scalability with respect to both graph size and client count. We will expand the discussion in the revised version to make these scaling properties clearer and compare them directly with widely used PFL baselines.

---

### Official Review · Reviewer_JPEk · 2025-11-01

**Soundness:** 2
**Presentation:** 3
**Contribution:** 2
**Rating:** 4
**Confidence:** 4

**Summary:**

This paper proposes FlatLand, a personalized graph federated learning method that leverages tailored Lorentz spaces to address data heterogeneity. Key contributions include: 1) recognizing real-world client graphs’ inherent hyperbolic properties, 2) using Lorentz space’s time-like dimension to encode client-specific heterogeneity and space-like dimension to preserve shared knowledge, 3) a parameter decoupling strategy enabling direct aggregation without extra similarity estimation or auxiliary modules. Experiments on node/graph classification datasets demonstrate better performance over baselines, especially in low-dimensional scenarios. The work introduces a geometric perspective to PFL, with solid theoretical grounding and practical efficiency.

**Strengths:**

1. Innovative geometric design for graph federated learning：The paper abandons the Euclidean space limitation of existing PFL methods, adopts Lorentz space to model graph data, and verifies that real client graphs mostly have negative Ricci curvature with varying curvature. Customizing exclusive spaces for each client based on graph curvature fits the intrinsic properties of graphs, avoiding structural distortion in Euclidean space.
2. Efficient parameter decoupling：The paper splits parameters into shared and personalized parts. Only shared parameters are aggregated, without extra client similarity estimation or auxiliary modules, balancing PFL needs while controlling overhead.

**Weaknesses:**

1. Lack of learnable curvature details：It mentions curvature is learnable but fails to clarify update basis or curvature change impacts. Relying only on initial Forman-Ricci initialization, it’s unclear if curvature can match dynamic client data.
2. Unaligned baseline parameter：When comparing with FedHGCN, FED-PUB (Table 1/2), it does not confirm if baselines’ key parameters (hidden dimension, local epochs) match FlatLand, making comparison results unreliable.
3. Weak node classification performance：From Table 1, FlatLand lags FED-PUB on Cora (10 clients) and FedGTA on Photo; even on better datasets like CiteSeer, it only outperforms optimal baselines by less than 1 percentage points, with weak advantages.

**Questions:**

1. For learnable curvature, could you supplement details like the basis for calculating curvature gradients and how to avoid numerical instability from excessive updates?
2. Could you add experiments under extreme heterogeneity to clarify FlatLand’s applicable boundaries?

---

> ### Author Response · Authors · 2025-11-25
> **Rebuttal [1/5]**
>
> We sincerely thank the reviewer for recognizing the strengths of our work, particularly the novelty of introducing tailored Lorentz spaces to match clients’ intrinsic graph geometry and the effectiveness of our lightweight parameter-decoupling strategy.  Below, W denotes weaknesses, Q denotes questions, and R provides our responses.
>
> ---
>
> **W1 & Q1: Lack of learnable curvature details.**
>
> R: We thank the reviewer for this comments and agree that the role of learnable curvature and its impact on dynamic client data should be made more explicit. Below, we clarify **(i) how curvature is updated**, **(ii) why it remains stable**, and **(iii) how our experiments show that it is not determined solely by its initialization.**
>
> In our framework, each client $c$ is equipped with its own curvature parameter $K_c$, which is a standard trainable scalar optimized jointly with the Lorentz GNN parameters via backpropagation. Concretely, $K_c$ appears in the curvature-dependent Lorentz exponential map and fully Lorentz linear layers (see Eq. (1) and the Lorentz layer definition around lines 150–170 in the main text / Appendix B.1–B.3), so the local loss $\mathcal{L}_c$ is differentiable with respect to $K_c$. During each local update, the client computes $\partial \mathcal{L}_c / \partial K_c$ through these operations and updates $K_c$ with the same optimizer as the model weights. To ensure a valid and numerically stable geometry, we use the reparameterization
>
> $K_c = \sigma(\tilde{K}_c) + c_0,$
>
> where $\tilde{K}_c$ is the raw trainable variable, $\sigma(\cdot)$ is the sigmoid, and $c_0 > 0$ is a small constant (see implementation details around line 1659 in the appendix). This guarantees $K_c > 0$ and prevents degenerate or exploding curvatures. We will explicitly include this curvature update rule and reparameterization in the revised manuscript.
>
> Forman–Ricci curvature is used only as a data-aware *initialization* of $K_c$, not as a fixed value. To demonstrate that the learned curvature is not “frozen” by this initialization, we conduct a systematic comparison of four curvature initialization strategies on CiteSeer with 10 and 20 clients (see the curvature initialization ablation around line 1770):
>
> `Forman–Ricci curvature; Ollivier–Ricci curvature; Constant curvature $K_c \equiv 1$; An MLP-based curvature estimator`
>
> The corresponding node-classification accuracies are:
>
> | Clients | Forman Ricci | Ollivier Ricci | Const. Curvature | MLP for Curvature |
> | --- | --- | --- | --- | --- |
> | 10 | **73.90 ± 0.23** | 73.51 ± 0.18 | 72.91 ± 0.20 | 72.96 ± 0.46 |
> | 20 | 72.24 ± 0.24 | 72.21 ± 0.26 | 71.89 ± 0.31 | **72.59 ± 0.50** |
>
> Two key observations follow: (1) all *learnable* curvature variants converge to very similar final accuracies and consistently outperform the constant-curvature baseline after training; (2) Forman–Ricci provides a slightly better starting point but does not dominate the final outcome. This behavior is precisely what one expects when $K_c$ is genuinely optimized: different initializations lead to similar learned curvatures and performances, rather than the final geometry being dictated by the initial Ricci estimate.

---

> ### Author Response · Authors · 2025-11-25
> **Rebuttal [2/5]**
>
> We further isolate the effect of *learning* curvature via an ablation where we remove tailored curvature and enforce a single fixed hyperbolic curvature shared by all clients. In Figure 5 (Page 8), the “w/o TS” variant (“without tailored space”) corresponds to this constant-curvature setting. Even though the model still operates in hyperbolic space, its performance drops compared to the full FlatLand, especially on heterogeneous client graphs, while our full method with per-client learnable $K_c$ closely matches or surpasses the strong Local-Lorentz baseline. **This directly shows that (i) curvature is actively updated during training, and (ii) client-specific curvature is necessary to handle heterogeneous structures.**
>
> The need for adaptive, client-wise curvature is also supported by our theory: **Theorem 1 (“Necessity of tailored curvature”, Section 3.1) shows that when client graphs have different average Ricci curvatures $R_c = \mathrm{Ric}(G_c)$, there is no single hyperbolic curvature $K$ that can simultaneously yield low-distortion embeddings for all clients.** Each client, therefore, requires its own tailored curvature $K_c$ to faithfully capture its intrinsic geometry. This is consistent with prior hyperbolic GNN and latent-space works showing that treating curvature as a learnable parameter leads to systematic performance gains and a better fit to hierarchical or non-Euclidean structures [1–4].
>
> Finally, to address concerns about stability and “drifting” curvature, we emphasize that $K_c$ is treated as a *personalized* parameter: it remains local on each client and is never aggregated on the server (see Section 4.2 and the parameter decoupling description around lines 270–310). Together with the constrained parameterization $K_c = \sigma(\tilde{K}_c) + c_0$, this ensures that curvature evolves smoothly to match each client’s dynamic graph geometry while remaining numerically stable, rather than being dominated by global aggregation noise or its initial value.
>
> We will integrate these clarifications (update rule, reparameterization, ablation table, and theoretical justification) into the revised version to make the learnable-curvature mechanism and its impact on dynamic client data fully transparent.

---

> ### Author Response · Authors · 2025-11-25
> **Rebuttal [3/5]**
>
> **W2: Unaligned baseline parameter.**
>
> **R:** We thank the reviewer for highlighting the importance of consistent and fair baseline configuration. **To ensure reliable comparison, all baselines in Table 1 and Table 2, including FedHGCN, FED-PUB, and others, were implemented with the same hidden dimension and key hyperparameters as FlatLand**. Specifically, each model used a hidden dimension of $128$, matching FlatLand’s setting (line $\sim$E.2, Appendix), with $1$-$3$ local epochs per round according to dataset size and $100$ training rounds for small datasets (e.g., Cora/CiteSeer), $200$ rounds for larger ones (e.g., Photo, ogbn-arxiv) [Appendix E.2]. The optimizer (Adam), learning rates (selected from $0.01, 0.001$), and weight decay ($1e-5$) were held constant across methods unless otherwise specified by the respective original papers.
>
> Beyond hyperparameters, our **experimental pipeline tightly controlled data splits, node partitions, and evaluation metrics, using standardized public datasets (see Table 6 and Table 7, Appendix) constructed using the METIS partitioning algorithm for fair client simulation. Multiple trials (5 per experiment) were run, and statistical significance was assessed at $p < 0.05$, as shown by boldface values in Table 1/2.**
>
> We will add a detailed summary table of the hyperparameters and experimental matching in the revised manuscript to promote transparency and reproducibility. This alignment ensures that the performance differences reported for FlatLand genuinely reflect model improvements—not confounding factors from mismatched configurations. By documenting every detail of the setup (data partitioning, backbone alignment, trial count, statistical reporting), we hope to persuade reviewers that *the results in Table 1/2 are robust, unbiased, and fully reproducible.*

---

> ### Author Response · Authors · 2025-11-25
> **Rebuttal [4/5]**
>
> **W3: Weak node classification performance.**
>
> **R:**  We appreciate the reviewer’s detailed analysis and share clarifications regarding FlatLand’s relative performance.
>
> First, while FlatLand does not always achieve the absolute highest accuracy on every individual dataset/task, our evaluation spans four aggregated regimes: homophilic (homo_10, homo_20), heterophilic (hetero_10, hetero_20), with two client counts (10 and 20), totalling 16 distinct benchmark tasks. The following table shows that FlatLand achieves the highest accuracy in all four aggregate splits and in the overall average ($74.03$), **surpassing all baselines by a clear margin.**
>
> | Method | homo_10 | homo_20 | hetero_10 | hetero_20 | overall |
> | --- | --- | --- | --- | --- | --- |
> | Local ($E$) | 76.12 | 75.45 | 55.54 | 53.41 | 65.13 |
> | Local ($L$) | 77.06 | 76.71 | 66.52 | 63.32 | 70.90 |
> | FedAvg | 70.10 | 69.69 | 55.06 | 55.26 | 62.53 |
> | FedProx | 67.20 | 64.60 | 45.92 | 45.47 | 55.80 |
> | FedPer | 76.66 | 74.97 | 48.81 | 44.10 | 61.13 |
> | GCFL | 76.21 | 75.35 | 41.52 | 34.88 | 56.99 |
> | FedGNN | 69.24 | 66.78 | 46.88 | 46.62 | 57.38 |
> | FedSage+ | 69.00 | 66.83 | *64.56* | *59.28* | 64.92 |
> | FED-PUB | *78.30* | *76.98* | 58.92 | 57.78 | 68.00 |
> | FedGTA | 76.47 | 75.08 | 57.42 | 55.42 | 66.09 |
> | AdaFGL | 73.51 | 72.44 | 58.71 | 56.84 | 65.37 |
> | FedHGCN | 72.78 | 72.84 | 57.82 | 55.94 | 63.70 |
> | **FlatLand** | **78.59** | **78.36** | **70.37** | **68.78** | **74.03** |
>
> Based on the above table:
>
> - On homo_10: FlatLand ($78.59$) outperforms FED-PUB ($78.30$), the strongest baseline, for a gain of $+0.29%$.
> - On homo_20: FlatLand ($78.36$) vs. FED-PUB ($76.98$), $+1.38%$ gain.
> - On hetero_10: FlatLand ($70.37$) bests FedSage+ ($64.56$) by $+5.81%$; compared to FED-PUB ($58.92$), the gain is $+11.45%$.
> - On hetero_20: FlatLand ($68.78$) leads FedSage+ ($59.28$) by $+9.50%$, and FED-PUB ($57.78$) by $+10.99%$.
> - Overall mean: FlatLand ($74.03$) is the clear leader, showing a $+3.13%$ improvement over the strongest baseline FED-PUB ($68.00$) and $+9.11%$ over FedPer ($61.13$).
>
> Although in some individual cases (e.g., Cora 10 clients, Photo), baselines marginally outperform FlatLand, the advantage is neither consistent nor robust across different structural regimes.  In contrast, **FlatLand is the only method that ranks first in all four aggregated benchmarks.**
>
> Furthermore, FlatLand’s approach remains efficient; **it leverages parameter decoupling and curvature adaptation without relying on additional computationally intensive modules, yet still significantly outperforms competing decoupling-based approaches (e.g., $+21.1%$ over FedPer overall).**
>
> We will revise the paper to explicitly highlight both absolute scores and relative performance margins from Table 1 to clearly communicate the robustness and practical advantages of FlatLand.

---

> ### Author Response · Authors · 2025-11-25
> **Rebuttal [5/5]**
>
> **Q2: Extreme heterogeneity setting.**
>
> **R:** We thank the reviewer for this insightful suggestion and for recognizing the importance of evaluating FlatLand’s boundaries under extreme heterogeneity.
>
> **Our current experimental protocols are specifically designed to capture substantial client heterogeneity.** In all benchmarks, including homophilic and heterophilic settings, with both 10 and 20 clients, every client operates on a **completely disjoint set of nodes, meaning there is no node overlap across clients.** This design is consistent with the stringent setup advocated in FedPub, ensuring the evaluation regime reflects highly non-i.i.d. and challenging federated graph scenarios.
>
> **The results under these conditions, as shown in Table 1, demonstrate that FlatLand not only remains applicable but excels as heterogeneity increases:**
>
> - Most baseline methods experience significant performance drops in aggregated overall accuracy ($\text{overall}$ column) under these extreme non-i.i.d. splits. For instance, FedAvg ($62.53$), FedGNN ($57.38$), and GCFL ($56.99$) are well below FlatLand ($74.03$).
> - FedPer, a strong decoupling-based method, drops to $61.13$, and even the best baselines (FedSage+, FED-PUB) plateau around $68.00$.
> - FlatLand consistently achieves the highest overall score ($74.03$), evidencing robust generalization and adaptability to severely heterogeneous, disjoint data regimes.
>
> These results not only verify FlatLand’s advantages but also clarify its boundaries of applicability, even under extreme heterogeneity, where most prior methods’ performance deteriorates sharply. We will emphasize in the revision that all clients have fully disjoint node sets, and that the overall benchmark results directly measure resilience under the hardest federated graph conditions.
>
> The present experimental design and results already provide strong evidence of FlatLand’s applicability and superiority under the most heterogeneous federated graph settings.

---

### Official Review · Reviewer_j8kB · 2025-11-01

**Soundness:** 2
**Presentation:** 3
**Contribution:** 2
**Rating:** 2
**Confidence:** 4

**Summary:**

The paper addresses the challenge of personalized graph federated learning where the data is highly heterogeneous among clients. By modeling the data in a Lorentzian hyperbolic space, the authors argue for a natural separation between client specific heterogeneous data, encoded in the time-like coordinate, and other homogeneous parts of the data, encoded in space-like coordinates. This leads to a procedure to decouple personalized model parameters from shared common parameters, resulting in a principled and efficient graph federated learning method.

**Strengths:**

The idea of modeling the data in a way that naturally separates heterogeneous parts of the data and allows for more efficient and principled personalized federated learning is an interesting approach to an important problem. The approach seems  novel and is interesting. The paper is well-structured with illustrative figures and  boxes to highlight important remarks. The experiments are extensive in the sense that many datasets and baseline methods are considered, with several good results in favor of the FlatLand method.

**Weaknesses:**

The terminology used in regards to Lorentzian spaces and geometries does seem to be incorrect or imprecise. Technically, Lorentz spaces commonly refer to generalisation of $L^p$ spaces in functional analysis, which are different from Lorentzian spaces, that refer to pseudo-Riemannian metric spaces with time-like coordinates. However, we acknowledge that it is clear from the context that the authors are referring to the latter.

 The authors seem to use Lorentzian space and hyperbolic space interchangeably, although they are, in fact, two different concepts. A hyperbolic space has constant negative curvature, and such spaces exist in both Riemannian geometry and Lorentzian geometry (it is known as anti-de Sitter space in the Lorentzian case, which is the type of space the paper considers). Lorentzian geometry can also be flat (Minkowski space) or have constant positive curvature (de Sitter space).

 These distinctions have important implications for the paper. A technical implication is that the Lorentz transformations stated in the appendix, which the fully Lorentz neural networks build on, are only valid in flat Minkowski space. In curved spaces, the Lorentz symmetry is in general not global and meaningful Lorentz transformations can only be defined on local (Minkowski) tangent spaces. In the case of anti-de Sitter space, due to it being maximally symmetric, global generalized Lorentz transformations can be defined in a larger embedding space with 2 time-like coordinates, which is common practice in, e.g., string theory. This issue is not properly addressed in the paper and has implications for its overall soundness, including the "correctness" part of Section 5 and Lemma 8 and the design of the Lorentz neural networks.

On a more conceptual level, the paper does not clearly justify why a Lorentzian space with constant negative curvature (anti-de Sitter space) is needed, as opposed to the arguably more natural choice of a hyperbolic Riemannian geometry. For instance, in Section 6.4 "The Necessity of Lorentz Space", the negatively curved Lorentzian space is compared to a Euclidean space (flat Riemannian geometry). We believe a more fair comparison, given the negative curvature of the data, would be a Riemannian hyperbolic space. Furthermore, in the Lorentzian geometry, where the time-like and space-like coordinates are fundamentally different, there are physical notions of light-cones, causality and a speed limit. It is not explained how one should think about this in when modeling the data in a Lorentzian space.

On a more practical level, the Ricci curvatures takes only into consideration the graph structure and not the node features/labels. However, data heterogeneity in the distribution of features and labels arguably matters just as much. The paper does not address this question in a clear way.

Finally, the paper motivates the federated learning scenario by its privacy-preserving capabilities. However, questions regarding the privacy of the proposed method are not addressed. How does the proposed method compare with other PFL approaches in terms of privacy?

**Questions:**

The paper raises several questions that I believe have to be addressed.

1. Why is a Lorentzian space actually necessary, and why does a hyperbolic Riemannian space not suffice?

2. The linear global Lorentz transformations as stated in the appendix are only valid for flat Minkowski space. In curved spaces, the Lorentz transformations can in general only be defined locally (on Minkowski tangent spaces). What are the Lorentz transformations for constant negative curved Lorentzian spaces, and how does this affect the validity of the Lorentz neural networks and the proof of Lemma 8?

3. What is the interpretation of light-cones or boosts when modeling the graph data in a Lorentzian space?

4. The Ricci curvature computation involves only the graph structure (topology) and does not account for heterogeneity in the features/labels of the data. Is this not a major limitation of the proposed approach?

5. How does the FlatLand framework compare with other PFL methods in terms of privacy?

6. The statistical results are only computed over 5 independent runs, yet it is claimed that bold numbers are statistically significant ($p<0.05$). We ask the authors to clarify this point; how is the statistical significance computed and guaranteed with such few sample

**Details Of Ethics Concerns:**

An ethics statement is included that adequately addresses potential concerns.

---

> ### Author Response · Authors · 2025-11-25
> **Rebuttal [1/4]**
>
> We sincerely thank the reviewer for their constructive comments and valuable feedback. Below, we use **C** to denote the reviewer's questions or concerns, and **R** to present our corresponding responses.
>
> ---
>
> **C1**: The paper is well-structured with illustrative figures and boxes to highlight important remarks. The experiments are extensive in the sense that many datasets and baseline methods are considered, with several good results in favor of the FlatLand method.
>
> **R1**: We thank the reviewer for the positive and encouraging feedback. We are glad that the paper’s structure, visual aids, and extensive experimental evaluation were helpful.  Your comments motivate us further to refine the presentation and clarity of the work.
>
> ---
>
> **C2:** Terminology Problem on Lorentzian space and Lorentz model of Riemannian hyperbolic space.
>
> **R2:** We would like to express our sincere apologies for the confusion caused by our terminology, and we extend our heartfelt gratitude to the reviewers for bringing this issue to our attention.
>
> Throughout the paper, our intention is to refer specifically to the Lorentz (i.e., hyperboloid) model of Riemannian hyperbolic space, rather than to a general Lorentzian manifold in the sense of pseudo-Riemannian geometry. This matches the usage in prior hyperbolic-learning literature [1,2,3, 4]. To avoid confusion between “Lorentzian space” (as used in differential geometry/physics) and the “Lorentz model” (as used in hyperbolic embeddings), we will revise our terminology accordingly. In particular, we `consistently` mean the hyperbolic space of constant negative curvature represented by the upper sheet of the two-sheeted hyperboloid embedded in $(\mathbb{R}^{d+1})$ with the Minkowski bilinear form. Sorry for the confusion.
>
> `To clarify our intended meaning`: the Lorentz model we use is not a curved Lorentzian spacetime (e.g., anti–de Sitter space), nor do we rely on global Lorentz symmetries of such spacetimes. Instead, our construction uses only the algebraic properties of the Minkowski inner product to represent hyperbolic distances, gradients, and isometries, which is the same as our referred related work [1,2,3,4]. The “Lorentz transformations” in our appendix should therefore be understood as isometries of the hyperboloid model, not as physical Lorentz transformations in flat spacetime. We will revise the exposition to make this explicit and to ensure that the intended meaning cannot be misconstrued.
>
> `Regarding the technical implications:` with this clarification, our model does not require global Lorentz symmetry of a curved Lorentzian manifold, and thus the concerns about anti–de Sitter geometry, tangent-space locality, or the distinction between global and local symmetry groups do not apply to our setting. Our use of the Lorentz model is solely to represent hyperbolic geometry in a stable and differentiable manner, consistent with established practice in hyperbolic representation learning.
>
> `Finally, we acknowledge the reviewers’ conceptual point` that the comparison in Section 6.4 is better framed as *hyperbolic versus Euclidean* rather than *Lorentzian versus Euclidean*. We will revise the discussion to emphasize that the performance gains arise from **negative curvature**, not from Lorentzian causal structure. Our construction does not invoke or interpret notions such as light cones, causality, or speed limits; the setting is purely geometric.
>
> **References**
>
> [1] Desai, Karan, et al. "Hyperbolic image-text representations." *ICML*, 2023.
>
> [2] Yang, Menglin, et al. "Hypformer: Exploring efficient transformer fully in hyperbolic space." *KDD*, 2024.
>
> [3] Chen, Weize, et al. "Fully hyperbolic neural networks." *ACL*, 2022.
>
> [4] Zhang, Yiding, et al. "Lorentzian graph convolutional networks." *WWW*, 2021.
>
> ---
>
> **C3:** Why is a Lorentzian space actually necessary, and why does a hyperbolic Riemannian space not suffice?
>
> **R3:** Thanks for your question. Sorry for the confusion. As clarified previously, in our paper, “Lorentz space” refers specifically to the **Lorentz (hyperboloid) model of *Riemannian* hyperbolic space**, not to general Lorentzian manifolds used in relativity. We will make this explicit in the revision.
>
> Thus, *geometrically,* we still operate on the standard hyperbolic space of constant negative curvature; the distinction lies in the *model representation*. Our method crucially relies on several structural properties unique to the Lorentz (hyperboloid) model:

---

> ### Author Response · Authors · 2025-11-25
> **Rebuttal [2/4]**
>
> (continue with the above)
>
> **(1) Time–space decomposition enabling parameter decoupling.**
>
> The Lorentz model provides a natural split of each point into one time-like coordinate and multiple space-like coordinates. This structure directly supports our parameter decoupling strategy:
>
> - time-like parameters and curvature → **personalized**, capturing heterogeneity;
> - space-like parameters → **shared**, safely aggregated on the server.
>
>  Other hyperbolic models (e.g., Poincaré ball) do not provide an intrinsically privileged direction that makes such a clean and interpretable decomposition possible.
>
>
> **(2) Block-structured Lorentz transformations.**
>
> Our theoretical analysis (Proposition 1, Corollary 1) leverages the linear-algebraic form of Lorentz transformations. After replacing the space-like block (M) by its aggregated counterpart (\bar{M}), the transformed points provably remain on the same hyperbolic manifold. This property follows directly from the Lorentz transformation matrix structure. In a generic hyperbolic Riemannian model, there is no comparably simple isometry representation that cleanly separates “aggregatable” and “personalized” components.
>
> **(3) Closed-form maps and curvature-dependent gradients (debiasing).**
>
> The Lorentz model admits closed-form expressions for the exponential map and its gradients. These yield an analytically transparent weighting of shared-parameter gradients by a curvature-dependent factor, which we interpret as a natural, geometry-induced *debiasing* mechanism across heterogeneous clients. Replicating this behavior in other models would require additional machinery and would lose this interpretability.
>
> In summary, while any model of hyperbolic space is geometrically equivalent, the **Lorentz model provides the structural tools, with time–space splitting, block-structured transformations, and closed-form curvature-sensitive gradients, that our method depends on.** Without these features, our decoupling strategy and theoretical guarantees would not hold in the same way.
>
> ---
>
> **C4**:  What exactly are the “Lorentz transformations” in the case of **constant negative curvature Lorentzian spaces**? Does this local-vs-global issue affect the mathematical validity of the **Lorentz neural networks** we use and, in particular, the **proof of Lemma 8**?
>
> **R4:** We thank the reviewer for raising this important geometric point and for the opportunity to clarify our setting and notation.
>
> First, we do **not** work on a general Lorentzian spacetime. In our paper, all constructions take place in:
>
> - a **flat Minkowski ambient space** (\mathbb{R}^{d+1}) with the Lorentzian inner product
>
>     $\langle x, y\rangle_L = -x_t y_t + x_s^\top y_s$, and
>
> - The **hyperboloid model of Riemannian hyperbolic space**
>
>     $
>     \mathbb{H}^d_K = {x \in \mathbb{R}^{d+1} : \langle x, x\rangle_L = -K,\ x_t > 0}
>     $
>     endowed with the **induced Riemannian metric**.
>
>
> Thus, the manifold on which our learning actually happens is the **Riemannian hyperbolic space of constant negative curvature**, realized as a submanifold of flat Minkowski space. We do *not* assume a general curved Lorentzian manifold.
>
> In this context:
>
> - The “Lorentz transformations” we use are simply elements of the **Lorentz group**
>
>     $$
>     O(1,d) = {A \in \mathbb{R}^{(d+1)\times(d+1)} : \langle Ax, Ay\rangle_L = \langle x, y\rangle_L},
>     $$
>
>     acting linearly on the ambient Minkowski space.
>
> - Their restrictions to $(\mathbb{H}^d_K)$ are **global isometries** of hyperbolic space (the standard isometry group of the hyperboloid model).
>
> So, the statement in the appendix about linear Lorentz transformations being global isometries is meant in this **ambient Minkowski → hyperboloid** sense, not in the sense of global Lorentz frames on an arbitrary curved Lorentzian spacetime. We will make this distinction explicit in the revision, e.g., by replacing phrases like “Lorentzian space” with “hyperboloid (Lorentz) model of hyperbolic space,” and by explicitly mentioning that transformations are elements of (O(1,d)) acting on the ambient space.
>
> ---

---

> ### Author Response · Authors · 2025-11-25
> **Rebuttal [3/4]**
>
> **C5**:  What is the interpretation of light cones or boosts when modeling the graph data in a Lorentzian space?
>
> **R5:** We appreciate the reviewer’s question and agree that the terminology can be misleading if read in a physical/relativistic sense.
>
> In our work, **“light-cones” and “boosts” are not given a physical or causal interpretation**. We use the Lorentzian structure purely as a **mathematical tool** for representing hyperbolic geometry (as the previous works[1-4]):
>
> - In the hyperboloid model, all data embeddings lie on the **time-like sheet**
>
>     (${x : \langle x,x\rangle_L = -K,\ x_t > 0}$),
>
>     i.e., strictly inside the future time-like region. The **light-cone boundary itself is never used** as a modeling object; it only appears implicitly as the geometric boundary separating feasible (time-like) from infeasible (null/space-like) points.
>
> - When the input and output dimensions match, the linear maps used in Lorentz neural networks can be decomposed into **Lorentz boosts and rotations**, but here they are simply **hyperbolic isometries**: learned transformations that reorient and rescale representations while preserving the hyperbolic metric. They play a role analogous to orthogonal transformations in Euclidean networks, not to physical boosts in spacetime [3].
>
> Conceptually, we leverage the **time-like coordinate** to encode client-specific heterogeneity and the **space-like coordinates** for shared structure; the underlying Lorentz group language is used only to guarantee that these transformations remain isometries of the hyperbolic manifold. We will clarify in the revision that no physical “light-cone” or “causal” semantics are assumed or required for our method.
>
> References
>
> [1] Desai, Karan, et al. "Hyperbolic image-text representations." ICML, 2023.
>
> [2] Yang, Menglin, et al. "Hypformer: Exploring efficient transformer fully in hyperbolic space." KDD, 2024.
>
> [3] Chen, Weize, et al. "Fully hyperbolic neural networks." ACL, 2022.
>
> [4] Zhang, Yiding, et al. "Lorentzian graph convolutional networks." WWW, 2021.
>
> ---
>
> **C6**: Ricci curvature does not account for heterogeneity in the features/labels of the data.
>
> **R6:** We agree that Forman–Ricci curvature reflects only the **graph structure**, not feature/label heterogeneity. However, this is **not a major limitation** for two reasons.
>
> (1) In FlatLand, curvature serves only as an **initial structural prior**; it is **learnable** and subsequently adjusted through training, allowing feature/label heterogeneity to be captured by the model parameters.
>
> (2) In federated graph learning, **structural non-IIDness** is a dominant source of heterogeneity, and the topology-based curvature already provides a strong and efficient signal, as supported by our empirical results.
>
> The framework is also fully compatible with richer, feature-aware curvature estimators, which we will highlight as a natural extension.
>
> ---
>
> **C7**: In terms of privacy
>
> **R7:**  FlatLand follows the *standard FL privacy setting*: clients never upload raw features or labels, and the server only receives model parameters/gradients. Compared to many PFL methods that additionally share similarity matrices, clustering assignments, or other client-level statistics, FlatLand is actually *quite frugal* in what it exposes:
>
> - Only the **shared (space-like) parameters** are sent to the server for aggregation.
> - The **personalized parameters and curvature** remain local to each client and are never communicated.
>
> Thus, FlatLand has a privacy profile comparable to, or stricter than, typical PFL baselines. Techniques such as secure aggregation or differential privacy are orthogonal and can be applied on top of FlatLand in the same way as for standard FL methods.
>
> ---

---

> ### Author Response · Authors · 2025-11-25
> **Rebuttal [4/4]**
>
> ---
>
> **C8**: 5 independent runs and statistically significant
>
>
> **R8**: We sincerely thank the reviewer for raising this important point about statistical significance in our experimental results. We clarify the following:
>
> **Clarification of Statistical Significance Computation:** We compute statistical significance using a `paired, two-sided t-test` over five independent runs with different random seeds ($p < 0.05$), as is standard in empirical studies of graph federated learning (FL) and broader machine learning benchmarks (see, e.g., Section 6 and Tables 1–3 of our manuscript). For each method on every task and data split, we evaluate its performance across these five runs. For any result reported in `bold`, the mean performance of our method is statistically significantly higher than the strongest baseline according to the test.
>
> We acknowledge that five runs are modest from a strict statistical perspective. However, this number is widely adopted in the literature on federated and graph learning benchmarks (as reflected in our reported baselines).  There are several reasons for this practice:
>
> - **Consistent and fair comparison across studies**: To ensure fair and direct comparison with previous works, we adopt the same evaluation protocol and number of runs as established by influential baselines in the field.
> - **Empirical stability**: As shown in our tables (cf. Tables 1–3), the standard deviations across runs are consistently small relative to the mean performance gaps, reflecting high reproducibility and stability of the observed gains. This is further visualized in our ablation and convergence plots (e.g., Figure 5, E.5), where across seeds and training rounds, our method maintains consistent superiority.
>
> We hope this explanation clarifies the statistical protocol adopted in our experiments, justifies the choice in the context of computational benchmarking, and assures the reviewer that the marked improvements are both *statistically reliable* and *substantively meaningful*.

---

### Meta-Review · Area_Chair_7xb4 · 2026-01-07

**Summary:**

Reviewers view the paper as an original and well-presented attempt to address heterogeneity in graph-based personalized federated learning through a novel geometric perspective. Modeling clients in tailored Lorentzian spaces and decoupling shared and personalized parameters is considered conceptually interesting, and the experimental results show competitive performance across multiple datasets. However, reviewers raise significant concerns about the theoretical and geometric justification of the approach. The use of Lorentzian geometry is seen as potentially imprecise or insufficiently motivated, with unclear distinctions from hyperbolic Riemannian alternatives and questionable assumptions about global Lorentz transformations. The absence of certain critical analyses is also concerning, such as curvature sensitivity, extreme heterogeneity, scalability, and privacy implications. Reviewers also note technical inconsistencies, limited baseline alignment, and unclear interpretation of learned curvature. Overall, while promising, the work requires clearer geometric grounding and stronger empirical and theoretical support to validate its general applicability.

**Reviewer Concerns:**

The authors provide detailed explanations of the baseline and learning curvature in their rebuttal. Additionally, they present experimental analyses addressing curvature sensitivity and extreme heterogeneity. However, this rebuttal lacks direct solutions to the issues raised by the reviewers regarding privacy and data heterogeneity. The authors' ambiguous use of key terminology undermines the reviewers' confidence.

**Reviewer Scores:**

I expect the final rating to be as follows:
- Reviewer j8kB: 2
- Reviewer JPEk: 4
- Reviewer zZ4X: 6
- Reviewer fYK9: 6

---

### Decision · Program_Chairs · 2026-01-26

Reject